# Log-Concave Sampling on Compact Supports: A Versatile Proximal Framework

## Abstract

In this paper, we investigate the theoretical aspects of sampling from strongly log-concave distributions defined on convex and compact supports. We propose a general proximal framework that involves projecting onto the constrained set, which is highly flexible and supports various projection options. Specifically, we consider the cases of Euclidean and Gauge projections, with the latter having the advantage of being performed efficiently using a membership oracle. This framework can be seamlessly integrated with multiple sampling methods. Our analysis focuses on Langevin-type sampling algorithms within the context of constrained sampling. We provide nonasymptotic upper bounds on the $W_1$ and $W_2$ errors, offering a detailed comparison of the performance of these methods in constrained sampling.

## 1 Introduction

Sampling from probability distributions plays a critical role in various fields of science and engineering, especially when dealing with convex and compact sets (Andrieu et al., 2003; Gelman et al., 1995; Stuart, 2010). In this context, the problem involves sampling from a probability measure $\nu$ on such sets, characterized by its density function

$$\nu(x) = \frac{e^{-U(x)}}{\int_{\mathbb{R}^p} e^{-U(s)} \mathrm{d}s},$$

Here, $U(x) = f(x) + \ell_{\mathcal{K}}(x)$, where $f(x)$ represents a potential function and $\ell_{\mathcal{K}}(x)$ is an indicator function ensuring $x$ lies within the convex and compact set $\mathcal{K} \subset \mathbb{R}^p$. Specifically, $\ell_{\mathcal{K}}(x)$ takes the form

$$\ell_{\mathcal{K}} := \begin{cases} +\infty & \text{if } x \notin \mathcal{K} \\ 0 & \text{if } x \in \mathcal{K}. \end{cases}$$

Solving this constrained sampling problem is challenging and has garnered considerable interest across various fields, including computer science and statistics. In the realm of computer science, a line of research initiated by Dyer et al. (1991) explored polynomial-time algorithms for uniformly sampling convex bodies. This has been followed by seminal studies on the convergence properties of the Ball Walk and the Hit-and-Run algorithm toward uniform density on a convex body or, more broadly, to log-concave densities (Kannan et al., 1997; Kook et al., 2024; Lovász, 1999; Lovász & Simonovits, 1993; Lovász & Vempala, 2007; Smith, 1984). Other Markov Chain Monte Carlo (MCMC) methods, such as Gibbs sampling (Gelfand et al., 1992) and Hamiltonian Monte Carlo (Brubaker et al., 2012; Gürbüzbalaban et al., 2022; Kook et al., 2022), have also been adapted and enhanced to sample from distributions defined on convex and compact sets.

In recent years, leveraging optimization techniques to facilitate sampling has become a prevalent approach. By formulating the sampling challenge as an optimization problem, methods like projected stochastic gradient descent (Bubeck et al., 2015; 2018; Lamperski, 2021; Lehec, 2023), proximal approaches (Brosse et al., 2017; Durmus et al., 2018; Salim & Richtárik, 2020), particle-based algorithms Li et al. (2022), and mirror descent (Ahn & Chewi, 2021; Chewi et al., 2020; Hsieh et al., 2018; Zhang et al., 2020) have proven effective in navigating the target distribution to generate samples. Further innovations have emerged from the intersection of deep learning and neural networks, leading to the development of novel sampling techniques via generative adversarial networks (Goodfellow et al., 2014) and variational autoencoders (Kingma & Welling, 2013). These advanced

methodologies offer promising pathways for sampling from intricate and high-dimensional distributions, particularly those defined on convex and compact sets (Ortiz-Haro et al., 2022). We will present a detailed discussion comparing our work with previous mentioned studies in Appendix A.

In this work, we tackle the challenge posed by a non-smooth target density $\nu$ by employing a proximal method. This method leverages a regularization technique that involves projecting onto set $\mathcal{K}$, effectively transforming the constrained sampling problem into an unconstrained one. Our framework is versatile, accommodating various projection options, such as the Euclidean projection—which corresponds to the Moreau envelope of the indicator function $\ell_{\mathcal{K}}$ (Rockafellar & Wets, 2009; Durmus et al., 2018; Brosse et al., 2017; Pereyra, 2016)—and the Gauge projection (Lu et al., 2022; Mhammedi, 2022), which can be efficiently performed using a membership oracle. Specifically, we introduce a smooth and strongly convex surrogate distribution that closely mimics the target density. By fine-tuning the regularization parameter, we can significantly reduce approximation errors. The advantageous properties of our surrogate distribution enable the effective application of various sampling techniques. In this work, we specifically explore the vanilla Langevin Monte Carlo (LMC) (Roberts & Tweedie, 1996; Dalalyan, 2017; Durmus & Moulines, 2017; Erdogdu & Hosseinzadeh, 2021; Mousavi-Hosseini et al., 2023; Raginsky et al., 2017; Erdogdu et al., 2018; Mou et al., 2022; Erdogdu et al., 2022), kinetic Langevin Monte Carlo (KLMC) (Cheng et al., 2018; Dalalyan & Riou-Durand, 2020; Shen & Lee, 2019; Ma et al., 2021; Zhang et al., 2023), and the parallelized versions of the midpoint randomization method for these algorithms (Shen & Lee, 2019; Yu & Dalalyan, 2024; He et al., 2020; Yu et al., 2023) (referred to as pRLMC and pRKLMC, respectively), within the context of constrained sampling. We derive the convergence rates for these algorithms in Wasserstein-1 and Wasserstein-2 distances, and provide a detailed comparison of their performance. To this end, we make the following contributions.

- We establish both general upper and lower bounds for the distance between the smooth approximation of the target density and its original form in Wasserstein-$q$ distance for any $q \geqslant 1$. These bounds are detailed in Proposition 2.1 and Proposition 2.2, respectively.
- In Section 2, we demonstrate that our proposed framework can seamlessly incorporate various projection options, including both Euclidean and Gauge projections.
- In Section 3, we incorporate several MCMC sampling methods, such as (kinetic) Langevin Monte Carlo and the parallelized midpoint randomization method for these algorithms, into our general framework. We present a detailed convergence analysis of these methods in both Wasserstein-1 and Wasserstein-2 distances.

In summary, we develop a comprehensive framework specifically designed for sampling from convex and compact sets, utilizing a regularization technique that involves projecting onto these constrained sets. Our study notably presents an improved error bound for LMC in constrained sampling settings, as well as the first convergence analysis for KLMC, pRLMC, and pRKLMC algorithms in this scenario. We emphasize that the convergence analysis for these three algorithms relies heavily on the smooth properties of the target density—conditions that are not met in the constrained setting we examine. Our proposed framework addresses these challenges and is exceptionally flexible and adaptable, accommodating various projection operators and sampling methods. Additionally, it enables clear comparisons of the behaviors of different sampling methods in constrained sampling scenarios. Overall, this new framework provides valuable theoretical insights into the dynamics of constrained sampling.

**Notation.** Denote the $p$-dimensional Euclidean space by $\mathbb{R}^p$. The letter $\boldsymbol{\theta}$ denotes the deterministic vector and its calligraphic counterpart $\boldsymbol{\vartheta}$ denotes the random vector. We use $\mathbf{I}_p$ and $\mathbf{0}_p$ to denote, respectively, the $p \times p$ identity and zero matrices. Define the relations $\mathbf{A} \preccurlyeq \mathbf{B}$ and $\mathbf{B} \succcurlyeq \mathbf{A}$ for two symmetric $p \times p$ matrices $\mathbf{A}$ and $\mathbf{B}$ to mean that $\mathbf{B} - \mathbf{A}$ is positive semi-definite. The gradient and the Hessian of a function $f : \mathbb{R}^p \to \mathbb{R}$ are denoted by $\nabla f$ and $\nabla^2 f$, respectively. Given any pair of measures $\mu$ and $\nu$ defined on $(\mathbb{R}^p, \mathcal{B}(\mathbb{R}^p))$, the Wasserstein-$q$ distance between $\mu$ and $\nu$ is defined as

$$\mathsf{W}_q(\mu, \nu) = \left( \inf_{\varrho \in \Gamma(\mu, \nu)} \int_{\mathbb{R}^p \times \mathbb{R}^p} \|\boldsymbol{\theta} - \boldsymbol{\theta}'\|_2^q \, \mathrm{d}\varrho(\boldsymbol{\theta}, \boldsymbol{\theta}') \right)^{1/q}, \quad q \geqslant 1 \,,$$

where the infimum is taken over all joint distributions $\varrho$ that have $\mu$ and $\nu$ as marginals. For a set $\mathcal{K} \subset \mathbb{R}^p$, we use $\mathcal{K}^c$ to denote its complement. The ceiling function maps $x \in \mathbb{R}$ to the smallest integer greater than or equal to $x$, denoted by $\lceil x \rceil$.

## 2 SMOOTH APPROXIMATION FOR THE TARGET DENSITY

The absence of smoothness in the target distribution $\nu$ presents significant challenges because sampling algorithms often depend on the smoothness properties of the target distribution to effectively explore the space and produce representative samples. Motivated by this, we consider the approximation for $\ell_{\mathcal{K}}$ of the form

$$\ell_{\mathcal{K}}^{\lambda}(\boldsymbol{\theta}) := \frac{1}{2\lambda^2} \|\boldsymbol{\theta} - \mathrm{P}_{\mathcal{K}}(\boldsymbol{\theta})\|_2^2 ,$$

where $\lambda > 0$ is the tuning parameter and $\mathrm{P}_{\mathcal{K}} : \mathbb{R}^p \to \mathcal{K}$ denotes a projection operator which projects the vector $x \in \mathbb{R}^p$ onto the set $\mathcal{K}$. Define $U^{\lambda}(\boldsymbol{\theta}) := f(\boldsymbol{\theta}) + \ell_{\mathcal{K}}^{\lambda}(\boldsymbol{\theta})$, and we the define the corresponding surrogate target density $\nu^{\lambda}$

$$\nu^{\lambda}(\boldsymbol{\theta}) = \frac{e^{-U^{\lambda}(\boldsymbol{\theta})}}{\int_{\mathbb{R}^p} e^{-U^{\lambda}(\boldsymbol{\theta}')} \mathrm{d}\boldsymbol{\theta}'} . \tag{2.1}$$

Throughout the paper, we define $\mu_k(\nu) = \int \|\boldsymbol{\theta}\|_2^k \, \nu(\mathrm{d}\boldsymbol{\theta})$ as the $k$-th moment of the distribution $\nu$ $(k \geqslant 1)$, and we assume the convex and compact set $\mathcal{K}$ satisfy the following assumption.

**Assumption 2.1.** *Given a positive constant $r \in (0, \infty)$, we assume the Euclidean ball centered at the origin with radius $r$, denoted by $\mathcal{B}_2(r)$, is contained in $\mathcal{K}$*

$$\mathcal{B}_2(r) = \{\boldsymbol{\theta} \in \mathbb{R}^p : \|\boldsymbol{\theta}\|_2 \leqslant r\} \subset \mathcal{K} .$$

This assumption has been commonly made in the work of constrained sampling (Lamperski, 2021; Bubeck et al., 2018; Brosse et al., 2017; Gürbüzbalaban et al., 2022). Moreover, we assume that the potential function $f$ satisfies the following assumption.

**Assumption 2.2.** *The function $f : \mathbb{R}^p \to \mathbb{R}$ is continuously differentiable, and its Hessian matrix $\nabla^2 f$ satisfies*

$$m\mathbf{I}_p \preccurlyeq \nabla^2 f(\boldsymbol{\theta}) \preccurlyeq M\mathbf{I}_p, \qquad \forall \boldsymbol{\theta} \in \mathbb{R}^p.$$

*Moreover, we assume the function $f$ is lower bounded and $\mathbf{0} = \arg\min_{\boldsymbol{\theta} \in \mathcal{K}} f(\boldsymbol{\theta})$.*

The assumption that $f$ attains the minimum at the origin simplifies the presentation of our results. All our results will still hold even if this condition is not met. Given the minimizer of the function $f$ over $\mathcal{K}$, denoted by $\boldsymbol{\theta}_*$, we simply need to shift all the coordinates and consider the constrained set $\mathcal{K} + \boldsymbol{\theta}_*$. Additionally, we note that the approximation $U^{\lambda}$ inherits the strong convexity of $f$. Moreover, we require $U^{\lambda}$ to be integrable, continuously differentiable, and smooth, even if $U$ is not.

**Condition 2.1.** *The function $U^{\lambda}$ is $m$-strongly convex, continuously differentiable, and $M^{\lambda}$-smooth. Moreover, it holds that $\int_{\mathbb{R}^p} e^{-U^{\lambda}(\boldsymbol{\theta})} \mathrm{d}\boldsymbol{\theta} < \infty$.*

Below, we present two examples of the approximation $\ell_{\mathcal{K}}^{\lambda}$, each utilizing a different, commonly used projection operator. We will demonstrate that these examples effectively approximate the indicator function $\ell_{\mathcal{K}}$ and the resulting potential functions fulfill Condition 2.1.

**Example 2.1.** *[Moreau envelope] One example of a projection operator is the Euclidean operator, which results in the Moreau envelope of the indicator function $\ell_{\mathcal{K}}$*

$$\ell_{\mathcal{K}}^{E,\lambda}(\boldsymbol{\theta}) = \inf_{\boldsymbol{\theta}' \in \mathbb{R}^p} \left( \ell_{\mathcal{K}}(\boldsymbol{\theta}') + \frac{1}{2\lambda^2} \|\boldsymbol{\theta} - \boldsymbol{\theta}'\|_2^2 \right) = \frac{1}{2\lambda^2} \|\boldsymbol{\theta} - \mathrm{P}_{\mathcal{K}}^E(\boldsymbol{\theta})\|_2^2 ,$$

*where $\mathrm{P}_{\mathcal{K}}^E$ denotes the Euclidean projection on $\mathcal{K}$.*

Define the corresponding surrogate potential $U^{E,\lambda}(\boldsymbol{\theta}) := f(\boldsymbol{\theta}) + \ell_{\mathcal{K}}^{E,\lambda}(\boldsymbol{\theta})$. This surrogate potential $U^{E,\lambda}$ satisfies the following property.

**Lemma 2.1.** *Assume Assumptions 2.1 and 2.2 hold, the potential $U^{E,\lambda}$ satisfies Condition 2.1 with $M^{\lambda} = 1/\lambda^2 + M$.*

**Example 2.2.** *[Gauge projection] Another example of the projection operator is the Gauge projection $\mathrm{P}_{\mathcal{K}}^G : \mathbb{R}^p \to \mathbb{R}$, which is defined as*

$$\mathrm{P}_{\mathcal{K}}^G(\boldsymbol{\theta}) := \frac{\boldsymbol{\theta}}{g_{\mathcal{K}}(\boldsymbol{\theta})}$$

*with a variation of Gauge function (also known as the Minkowski function) $g_{\mathcal{K}} : \mathbb{R}^p \to \mathbb{R}$ of set $\mathcal{K}$*

$$g_{\mathcal{K}}(\boldsymbol{\theta}) := \inf\{t \geqslant 1 : \boldsymbol{\theta} \in t\mathcal{K}\} . \tag{2.2}$$

*This function encapsulates the scaling required to project $\boldsymbol{\theta}$ onto the set $\mathcal{K}$.*

Set $\ell_{\mathcal{K}}^{G,\lambda}(\boldsymbol{\theta}) = \frac{1}{2\lambda^2}\|\boldsymbol{\theta} - \mathrm{P}_{\mathcal{K}}^G(\boldsymbol{\theta})\|_2^2$. The corresponding surrogate function is defined via $U^{G,\lambda}(\boldsymbol{\theta}) :=$ $f(\boldsymbol{\theta}) + \ell_{\mathcal{K}}^{G,\lambda}(\boldsymbol{\theta})$. This surrogate potential $U^{G,\lambda}$ satisfies the following property.

**Lemma 2.2.** *Assume Assumptions 2.1 and 2.2 hold, the surrogate potential $U^{G,\lambda}$ satisfies Condition 2.1 with $M^\lambda = 1/\lambda^2 + M$.*

In the following, we aim to measure the discrepancy between the surrogate distribution and the target distribution. To this end, we establish an upper bound in $\mathsf{W}_q$ distance between the target distribution $\nu$ and the approximated distribution $\nu^\lambda$ defined in equation 2.1. Our aim is to demonstrate that as $\lambda$ approaches zero, the distance between these two distributions converges to zero.

**Proposition 2.1.** *Under Assumption 2.1, assume that the potential function $f$ is convex with $\mathbf{0}$ as its minimizer. Then, for any $q \geqslant 1$ and any $\lambda \in \left(0, \frac{r}{2(p+q)}\right)$, it holds that*

$$\mathsf{W}_q^q(\nu, \nu^\lambda) \leqslant 3\mu_q(\nu)\lambda\left(2p + q\right)/r\,,$$

*where $\mu_q(\nu) = \int \|\boldsymbol{\theta}\|_2^q\,\nu(\mathrm{d}\boldsymbol{\theta}), q \geqslant 1$.*

In this proposition, we provide a general and precise quantification of the distance between the approximate distribution $\nu^\lambda$ to the target distribution $\nu$ in $\mathsf{W}_q$ distance for any $q \geqslant 1$. When $q = 1$, our result aligns with the bound established for $\mathsf{W}_1(\nu, \nu^\lambda)$ as stated in Proposition 5 of Brosse et al. (2017). Notably, our Assumption 2.1 is less restrictive than the assumption outlined in Proposition 5 of that work, as we do not require the domain to be contained within a ball, as specified in Assumption H2 of that paper. In Proposition 2.1, the size of $\mathcal{K}$ is captured in the bounds through the radius $r$ and the $q$-th moment of the target distribution $\mu_q$. In this context, $\mu_q$ plays a role analogous to the radius of the ball containing $\mathcal{K}$, denoted as $R$ in Brosse et al. (2017), while offering a more precise description of the domain's size.

We also derive the lower bound for the distance between the approximate distribution $\nu^\lambda$ to the target distribution $\nu$ in $\mathsf{W}_q$ distance, specifically for the scenario where $\mathcal{K} = \mathcal{B}_2(r)$ and the projections discussed in Examples 2.1 and 2.2. In this case, the Gauge projection and Euclidean projection coincide, yielding the following results.

**Proposition 2.2.** *Let $\mathcal{K} = \mathcal{B}_2(r)$ and assume the potential function $f$ is convex and $M$-smooth, with $\mathbf{0}$ as its minimizer and $f(\mathbf{0}) = 0$. Set $\ell_{\mathcal{K}} = \ell_{\mathcal{K}}^{E,\lambda}$. Under Assumptions 2.1, for any $q \geqslant 1$ and any $\lambda \in \left(0, \frac{r}{2(p+q)} \wedge \frac{1}{\sqrt{M}}\right)$, it holds that*

$$\mathsf{W}_q^q(\nu, \nu^\lambda) \geqslant \mu_q(\nu)\min\left\{\left|\left(\sqrt{\tfrac{\pi}{8}}e^{-\frac{3}{4}Mr^2}\tfrac{\lambda(p+q)}{r+3\lambda p} + \tfrac{r}{r+3\lambda p}\right)^{1/q} - 1\right|^q, \left|\left(\tfrac{3\lambda(p+q)}{r} + 1\right)^{1/q} - 1\right|^q\right\}.$$

This result provides a quantitative assessment of the tightness of the upper bound established in Proposition 2.1. Before we discuss the results of the lower bound, we present the following corollary.

**Corollary 2.1.** *Under the assumptions stated in Proposition 2.2, when $\lambda = o(r/p)$ and $q = 1$, it holds that*

$$\mathsf{W}_1(\nu, \nu^\lambda) \geqslant C_{M,r}\mu_1(\nu)\tfrac{p+1}{r}\lambda\,,$$

*where $C_{M,r} > 0$ is a constant that depends on $M$ and $r$ exponentially.*

Based on the results outlined in the preceding corollary, when $q = 1$ and with a sufficiently small $\lambda$, the lower bound aligns with the rate estimates from Proposition 2.1 up to a constant factor, thereby confirming the optimality of the upper bound for the case of $q = 1$. However, when $q > 1$ and $\lambda$ is small, the lower bound follows the order $\left((1 + \lambda)^{1/q} - 1\right)^q$, which increases at a rate slower than $\lambda$. This indicates the suboptimality[1] of the rate stated in Proposition 2.1, highlighting a difference in behavior for higher values of $q$.

# 3 PROXIMAL LANGEVIN ALGORITHMS

A central theme of this work is the approximation of the non-smooth potential $\nu$ using a carefully crafted smooth surrogate density $\nu^\lambda$. We note that the convergence analysis for the log-concave

---

[1] We conjecture that this suboptimality arises from the general technique used in our proof. With specific assumptions about the density, a refined lower bound for the Wasserstein distance between $\nu$ and $\nu^\lambda$.

sampling algorithms typically relies heavily on the strongly log-concave and smooth properties of the target density. However, these conditions are not met in the constrained setting considered in this work. Owing to the desirable properties of the surrogate density $\nu^\lambda$, we now can utilize various sampling methods that are effective for distributions with smooth and strongly convex densities.

In this section, we demonstrate how the proposed scheme can be adapted to various sampling methods that collectively aim to sample from the density $\nu^\lambda$. We particularly focus on Langevin Monte Carlo (LMC) and its variant, Kinetic Langevin Monte Carlo (KLMC). LMC employs stochastic differential equations to sample effectively from complex distributions, making it a powerful tool in high-dimensional spaces. KLMC enhances efficiency by incorporating kinetic energy dynamics. Additionally, we explore the parallelized randomized midpoint method for these two algorithms, which improves sampling speed by leveraging parallel computations. Due to space constraints, we provide only a brief overview of each algorithm for the convenience of readers in the following subsection. We refer interested readers to the works of Dalalyan (2017); Durmus & Moulines (2017); Shen & Lee (2019); Yu & Dalalyan (2024); He et al. (2020); Yu et al. (2023) for further details on these algorithms.

More specifically, we combine the discretization error with the approximation error analyzed in Section 2 to evaluate errors in Wasserstein-1 and Wasserstein-2 distances between the sample distribution and the target density $\nu$ in the context of constrained sampling. Table 1 below offers a comparison of the results for the four sampling methods mentioned above, using the specific projection operators discussed in Example 2.1 and Example 2.2[2]. For simplicity, we omit the constants and logarithmic terms that appear in the bounds. The table illustrates that when the number of parallel steps $R > 1$, the randomized midpoint method markedly enhances the performance of both the vanilla Langevin Monte Carlo and kinetic Langevin Monte Carlo algorithms. When the number of parallel steps $R = 1$, the pRLMC and pRKLMC algorithms correspond exactly to the randomized midpoint method applied to Langevin Monte Carlo (RLMC) and kinetic Langevin Monte Carlo (RKLMC), respectively, as detailed in Yu & Dalalyan (2024). The convergence rate for RLMC is comparable to that of LMC when assessing the error in $W_2$ distance. Further details are provided following Corollary 3.2.

| | ‖ | LMC | KLMC | pRLMC | pRKLMC |
|---|---|---|---|---|---|
| $W_1$ | ‖ | $\widetilde{\mathcal{O}}(\varepsilon^{-4})$ | $\widetilde{\mathcal{O}}(\varepsilon^{-4})$ | $\widetilde{\mathcal{O}}(R^{-1/3}\varepsilon^{-10/3})$ | $\widetilde{\mathcal{O}}(R^{-1/3}\varepsilon^{-8/3})$ |
| $W_2$ | ‖ | $\widetilde{\mathcal{O}}(\varepsilon^{-6})$ | $\widetilde{\mathcal{O}}(\varepsilon^{-7})$ | $\widetilde{\mathcal{O}}(R^{-1/3}\varepsilon^{-6})$ | $\widetilde{\mathcal{O}}(R^{-1/3}\varepsilon^{-5})$ |

Table 1: The number of iterations required by {L,KL,pRL,pRKL}MC algorithms to achieve an error bounded by $\varepsilon\sqrt{p/m}$ in $W_1$ distance and $W_2$ distance. $R \geqslant 1$ denotes the number of parallel steps.

## 3.1 Langevin Monte Carlo (LMC)

Let $\boldsymbol{\vartheta}_0$ be a random vector drawn from a distribution $\nu^\lambda$ on $\mathbb{R}^p$ and let $\boldsymbol{W} = (\boldsymbol{W}_t : t \geqslant 0)$ be a $p$-dimensional Brownian motion that is independent of $\boldsymbol{\vartheta}_0$. To sample from the approximation distribution $\nu^\lambda$, we consider the vanilla Langevin diffusion, which is a strong solution to the stochastic differential equation

$$\mathrm{d}\boldsymbol{L}_t^{\mathsf{LD}} = -\nabla U^\lambda(\boldsymbol{L}_t^{\mathsf{LD}})\,\mathrm{d}t + \sqrt{2}\,\mathrm{d}\boldsymbol{W}_t, \qquad t \geqslant 0, \qquad \boldsymbol{L}_0^{\mathsf{LD}} = \boldsymbol{\vartheta}_0. \qquad (3.1)$$

This equation has a unique strong solution, which is a continuous-time Markov process, termed Langevin diffusion. Under the further assumptions on the potential $U^\lambda$, such as strong convexity, the Langevin diffusion is ergodic, geometrically mixing and has $\nu^\lambda$ as its unique invariant distribution (Bhattacharya, 1978). Moreover, we can sample from the distribution defined by $\nu^\lambda$ by using a suitable discretization of the Langevin diffusion. LMC algorithm is based on this idea, combining the considerations with the Euler discretization. Specifically, for small values of $h \geqslant 0$ and

---

[2]In this work, we mainly focus on the Wasserstein metric. To our knowledge, the only comparable results are those provided by Brosse et al. (2017) concerning the application of Langevin Monte Carlo. We offer a detailed discussion of this comparison following Corollary 3.1.

$\Delta_h \boldsymbol{W}_t = \boldsymbol{W}_{t+h} - \boldsymbol{W}_t$, the following approximation holds

$$\boldsymbol{L}_{t+h}^{\mathsf{LD}} = \boldsymbol{L}_t^{\mathsf{LD}} - \int_0^h \nabla U^\lambda(\boldsymbol{L}_{t+s}^{\mathsf{LD}}) \, \mathrm{d}s + \sqrt{2} \, \Delta_h \boldsymbol{W}_t \approx \boldsymbol{L}_t^{\mathsf{LD}} - h\nabla U^\lambda(\boldsymbol{L}_t^{\mathsf{LD}}) + \sqrt{2} \, \Delta_h \boldsymbol{W}_t.$$

By repeatedly applying this approximation with a small step-size $h$, we can construct a Markov chain $(\boldsymbol{\vartheta}_k^{\mathsf{LMC}} : k \in \mathbb{N})$ that converges to the target distribution $\nu^\lambda$ as $h$ goes to zero. More precisely, $\boldsymbol{\vartheta}_k^{\mathsf{LMC}} \approx \boldsymbol{L}_{kh}^{\mathsf{LD}}$, for $k \in \mathbb{N}$, is given by

$$\boldsymbol{\vartheta}_{k+1}^{\mathsf{LMC}} = \boldsymbol{\vartheta}_k^{\mathsf{LMC}} - h\nabla U^\lambda(\boldsymbol{\vartheta}_k^{\mathsf{LMC}}) + \sqrt{2} \, (\boldsymbol{W}_{(k+1)h} - \boldsymbol{W}_{kh}).$$

We now provide explicit upper bounds for the error of the LMC algorithm in terms of $\mathsf{W}_1$ and $\mathsf{W}_2$ distances within the context of constrained sampling.

**Theorem 3.1.** *Under Assumptions 2.1 and 2.2, we further assume that the potential $U^\lambda$ satisfies Condition 2.1. Let the step size $h \in (0, 1/M^\lambda)$ and the tuning parameter $\lambda \in \left(0, \frac{r}{2(p+2)}\right)$. Then, for every $n \geqslant 1$, the distribution $\nu_n^{\mathsf{LMC}}$ of $\boldsymbol{\vartheta}_n^{\mathsf{LMC}}$ satisfies*

$$\mathsf{W}_2(\nu_n^{\mathsf{LMC}}, \nu) \leqslant e^{-\frac{mnh}{2}} \mathsf{W}_2(\nu_0^{\mathsf{LMC}}, \nu) + 2\sqrt{\frac{3\mu_2(\nu)\lambda(2p+2)}{r}} + \sqrt{\frac{2phM^\lambda}{m}} \, .$$

*Moreover, when the initial distribution $\nu_0^{\mathsf{LMC}}$ is set to be the Dirac measure at the minimizer of $f$, it holds that*

$$\mathsf{W}_1(\nu_n^{\mathsf{LMC}}, \nu) \leqslant e^{-\frac{mnh}{2}} \sqrt{\frac{p}{m}} + \frac{3\mu_1(\nu)\lambda(2p+1)}{r} + \sqrt{\frac{2phM^\lambda}{m}} \, .$$

The term $M^\lambda$ introduces an additional factor of $\lambda$ into the convergence rate. To clarify this dependence on $\lambda$, we specify in the following corollary that $M^\lambda = M + 1/\lambda^2$. This specification corresponds to the specific projection operators used in Example 2.1 and Example 2.2. We then optimize $\lambda$ to obtain the results presented below.

**Corollary 3.1.** *Let $\varepsilon \in (0, 1)$ be a small number, and $M^\lambda = M + \frac{1}{\lambda^2}$.*
*(a) Set $\lambda = \sqrt{2/3}(ph/m)^{1/4}\sqrt{r/(\mu_1(\nu)(2p+1))}$, choose $h > 0$ and $n \in \mathbb{N}$ so that*

$$h = 2^{-10}3^{-2}(p/m)r^2\big(\mu_1(\nu)(2p+1)\big)^{-2}\varepsilon^4 \quad and \quad n \geqslant \frac{2}{mh}\log(2/\varepsilon) \, ,$$

*then we have $\mathsf{W}_1(\nu_n^{\mathsf{LMC}}, \nu) \leqslant \varepsilon\sqrt{p/m}$.*
*(b) Set $\lambda = 2^{-2/3}3^{-1/3}(rh)^{1/3}(\mu_2(\nu)m)^{-1/3}$, choose $h > 0$ and $n \in \mathbb{N}$ so that*

$$h = 2^{-20}3^{-2}r^2(\mu_2(\nu)m)^{-2}\varepsilon^6 \quad and \quad n \geqslant \frac{2}{mh}\log(2/\varepsilon) \, ,$$

*then we have $\mathsf{W}_2(\nu_n^{\mathsf{LMC}}, \nu) \leqslant \varepsilon\sqrt{p/m}$.*

According to this corollary, when evaluating the error in $\mathsf{W}_1$ distance, the required sample size $n$ to achieve a prespecified error level is of order $\tilde{\mathcal{O}}(\varepsilon^{-4})$. This represents an improvement over the rate of $\tilde{\mathcal{O}}(\varepsilon^{-6})$ obtained in Brosse et al. (2017). Additionally, our general framework provides the convergence rate in $\mathsf{W}_2$ for the proximal LMC algorithm, thus complementing the study of the proximal LMC presented in Brosse et al. (2017).

### 3.2 PARALLELIZED RANDOMIZED MIDPOINT DISCRETIZATION OF VANILLA LANGEVIN DIFFUSION (PRLMC)

As an alternative to the Euler discretization method discussed in Section 3.1 for the stochastic differential equation 3.1, we consider the randomized midpoint method, initially introduced in Shen & Lee (2019), as a different discretization framework. This method enables the discretization and simulation of the Langevin diffusion 3.1, ultimately converging to the distribution $\nu^\lambda$. Building upon the foundations laid by Shen & Lee (2019); Yu et al. (2023), Yu & Dalalyan (2024) developed a parallel computing scheme for this algorithm, significantly enhancing the efficiency of the sampling process. The formal definition of pRLMC is defined in Algorithm 1. For simplicity, the superscript pRLMC is omitted therein. We now introduce the primary findings in the work, which establish explicit upper bounds for the error associated with the pRLMC algorithm, measured in $\mathsf{W}_1$ and $\mathsf{W}_2$ distances, specifically within the framework of constrained sampling.

---

**Algorithm 1** Parallelized RLMC (pRLMC)

---

**Input**: number of parallel steps $R$, number of sequential iterations $Q$, step size $h$, number of iterations $K$, initial point $\boldsymbol{\vartheta}_0$.
**Output**: iterate $\boldsymbol{\vartheta}_{K+1}$

1: **for** $k = 1$ to $K$ **do**
2:     Draw $U_{kr}$ uniformly from $\left[\frac{r-1}{R}, \frac{r}{R}\right]$, $r = 1, \ldots, R$.
3:     Generate $\boldsymbol{\xi}_{kr} = \boldsymbol{W}_{(k+U_{kr})h} - \boldsymbol{W}_{kh}$, $r = 1, \ldots, R$.
4:     Set $\boldsymbol{\vartheta}_k^{(0,r)} = \boldsymbol{\vartheta}_k$, $r = 1, \ldots, R$.
5:     **for** $q = 1$ to $Q - 1$ **do**
6:         **for** $r = 1$ to $R$ in parallel **do**
7:             $a_{kj} = \min\{\frac{1}{R}, U_{kr} - \frac{j-1}{R}\}$, $j = 1, \ldots, r$
8:             $\boldsymbol{\vartheta}_k^{(q,r)} = \boldsymbol{\vartheta}_k - h \sum_{j=1}^r a_{kj} \nabla U^\lambda\big(\boldsymbol{\vartheta}_k^{(q-1,j)}\big) + \sqrt{2}\boldsymbol{\xi}_{kr}$.      In parallel
9:         **end for**
10:     **end for**
11:     $\boldsymbol{\vartheta}_{k+1} = \boldsymbol{\vartheta}_k - \frac{h}{R} \sum_{r=1}^R \nabla U^\lambda\big(\boldsymbol{\vartheta}_k^{(Q-1,r)}\big) + \sqrt{2}(\boldsymbol{W}_{(k+1)h} - \boldsymbol{W}_{kh})$.
12: **end for**

---

**Theorem 3.2.** *Under Assumptions 2.1 and 2.2, we further assume that the potential $U^\lambda$ satisfies Condition 2.1. Let the the tuning parameter $\lambda \in \left(0, \frac{r}{2(p+2)}\right)$. Choose the step size $h$ such that $M^\lambda h \leqslant 1/10$. Then, for every $n \geqslant 1$, the distribution $\nu_n^{\mathsf{pRLMC}}$ of $\boldsymbol{\vartheta}_n^{\mathsf{pRLMC}}$ satisfies*

$$
\mathsf{W}_2(\nu_n^{\mathsf{pRLMC}}, \nu) \leqslant \left(1 + \sqrt{\kappa M^\lambda h}\big(0.82(M^\lambda h)^{Q-1} + 0.94 M^\lambda h/R\big)\right) e^{-mnh/2} \mathsf{W}_2(\nu_0^{\mathsf{pRLMC}}, \nu)
$$

$$
+ \sqrt{\kappa M^\lambda h}\big(3.98(M^\lambda h)^{Q-1} + 6.91 M^\lambda h/\sqrt{R}\big)\sqrt{p/m}
$$

$$
\left(2 + \sqrt{\kappa M^\lambda h}\big(0.82(M^\lambda h)^{Q-1} + 0.94 M^\lambda h/R\big)\right)\sqrt{3\mu_2(\nu)\lambda(2p+2)/r} \,,
$$

*where $\kappa = M^\lambda/m$. Moreover, when the initial distribution $\nu_0^{\mathsf{pRLMC}}$ is set to be the Dirac measure at the minimizer of $f$, it holds that*

$$
\mathsf{W}_1(\nu_n^{\mathsf{pRLMC}}, \nu) \leqslant \left(1 + \sqrt{\kappa M^\lambda h}\big(0.82(M^\lambda h)^{Q-1} + 0.94 M^\lambda h/R\big)\right) e^{-mnh/2} \sqrt{p/m}
$$

$$
+ \sqrt{\kappa M^\lambda h}\big(3.98(M^\lambda h)^{Q-1} + 6.91 M^\lambda h/\sqrt{R}\big)\sqrt{p/m}
$$

$$
+ 3\mu_1(\nu)\lambda(2p+1)/r.
$$

To our knowledge, this represents the first convergence rate for the pRLMC algorithm within the context of constrained sampling. When $R = 1$, these results recover the convergence rate for the randomized midpoint method applied to Langevin Monte Carlo algorithm (RLMC) (He et al., 2020; Yu et al., 2023; Yu & Dalalyan, 2024) in constrained sampling. Below, we define $M^\lambda = M + 1/\lambda^2$, a setting that corresponds to the projection choices outlined in Examples 2.1 and 2.2. We then optimize $\lambda$ to derive the following corollary.

**Corollary 3.2.** *Given the number of parallel steps $R$, we set $Q = \lceil \log R \rceil + 1$. Let $\varepsilon \in (0, 1)$ be a small number, and $M^\lambda = M + \frac{1}{\lambda^2}$.*

*(a) Set $\lambda = \left(\frac{2^4 \cdot 7}{3}\right)^{1/5}\left(\frac{p}{R}\right)^{1/10}\left(\frac{r}{m\mu_1(\nu)(2p+1)}\right)^{1/5} h^{3/10}$, choose $h > 0$ and $n \in \mathbb{N}$ so that*

$$
h = 15.6^{-10/3} p^{2/5} m^{-1}\left(\frac{r}{\mu_1(\nu)(2p+1)}\right)^{8/3} R^{1/3} \varepsilon^{10/3} \quad and \quad n \geqslant \frac{2}{mh}\log\left(\frac{2.22}{\varepsilon}\right),
$$

*then we have $\mathsf{W}_1(\nu_n^{\mathsf{pRLMC}}, \nu) \leqslant \varepsilon\sqrt{p/m}$.*

*(b) $\lambda = 320^{2/9} 3^{-1/3}\left(\frac{rp}{m^2\mu_2(\nu)(2p+2)R}\right)^{1/9} h^{1/3}$, choose $h > 0$ and $n \in \mathbb{N}$ so that*

$$
h = 13^{-6} m^{-7/3}\left(\frac{rp}{\mu_2(\nu)(2p+2)}\right)^{8/3} R^{1/3} \varepsilon^6 \quad and \quad n \geqslant \frac{2}{mh}\log\left(\frac{2.22}{\varepsilon}\right),
$$

*then we have $\mathsf{W}_2(\nu_n^{\mathsf{pRLMC}}, \nu) \leqslant \varepsilon\sqrt{p/m}$.*

Comparing this convergence rate with that obtained for proximal LMC in Corollary 3.1, we observe superior performance of pRLMC over LMC when $R > 1$. When assessing the error in $\mathsf{W}_2$ distance for the case of $R = 1$, the sample complexity $\tilde{\mathcal{O}}(\epsilon^{-6})$ aligns with that of LMC. This outcome results from our specific choice of $\lambda$ during the optimization of the upper bound concerning $\lambda$, placing it within a region where the performance of LMC is comparable to that of RLMC.

## 3.3 KINETIC LANGEVIN MONTE CARLO (KLMC)

In this part, we explore the application of kinetic Langevin diffusion in constrained sampling. Recall that the kinetic Langevin process $\boldsymbol{L}^{\mathsf{KLD}}$ is a solution to a second-order stochastic differential equation that can be informally written as

$$\frac{1}{\gamma}\ddot{\boldsymbol{L}}_t^{\mathsf{KLD}} + \dot{\boldsymbol{L}}_t^{\mathsf{KLD}} = -\nabla U^\lambda(\boldsymbol{L}_t^{\mathsf{KLD}}) + \sqrt{2}\,\dot{\boldsymbol{W}}_t, \tag{3.2}$$

with initial conditions $\boldsymbol{L}_0^{\mathsf{KLD}} = \boldsymbol{\vartheta}_0$ and $\dot{\boldsymbol{L}}_0^{\mathsf{KLD}} = \mathbf{v}_0$. In equation 3.2, $\gamma > 0$, $\boldsymbol{W}$ is a standard $p$-dimensional Brownian motion and dots are used to designate derivatives with respect to time $t \geqslant 0$. This can be formalized using Itô's calculus and introducing the velocity field $\mathbf{V}^{\mathsf{KLD}}$ so that the joint process $(\boldsymbol{L}^{\mathsf{KLD}}, \mathbf{V}^{\mathsf{KLD}})$ satisfies

$$\mathrm{d}\boldsymbol{L}_t^{\mathsf{KLD}} = \mathbf{V}_t^{\mathsf{KLD}}\,\mathrm{d}t; \quad \frac{1}{\gamma}\mathrm{d}\mathbf{V}_t^{\mathsf{KLD}} = -\big(\mathbf{V}_t^{\mathsf{KLD}} + \nabla U^\lambda(\boldsymbol{L}_t^{\mathsf{KLD}})\big)\,\mathrm{d}t + \sqrt{2}\,\mathrm{d}\boldsymbol{W}_t. \tag{3.3}$$

Similar to the vanilla Langevin diffusion 3.1, the kinetic Langevin diffusion $(\boldsymbol{L}^{\mathsf{KLD}}, \mathbf{V}^{\mathsf{KLD}})$ is a Markov process that exhibits ergodic properties when the potential $U^\lambda$ is strongly convex (see Eberle et al. (2019) and references therein). The invariant density of this process is given by

$$p_*(\boldsymbol{\theta}, \mathbf{v}) \propto \exp\{-U^\lambda(\boldsymbol{\theta}) - \tfrac{1}{2\gamma}\|\mathbf{v}\|^2\}, \qquad \text{for all} \quad \boldsymbol{\theta}, \mathbf{v} \in \mathbb{R}^p.$$

Note that the marginal of $p_*$ corresponds to $\boldsymbol{\theta}$ coincides with the target density $\nu^\lambda$. The kinetic Langevin Monte Carlo (KLMC) algorithm is a discretized version of KLD 3.3, where the term $\nabla U^\lambda(\boldsymbol{L}_t)$ is replaced by $\nabla U^\lambda(\boldsymbol{L}_{kh})$ on each interval $[kh, (k+1)h)$. The resulting error bounds are given in the following theorem.

**Theorem 3.3.** *Under Assumptions 2.1 and 2.2, we further assume that the potential $U^\lambda$ satisfies Condition 2.1. Let the tuning parameter $\lambda \in \big(0, \frac{r}{2(p+2)}\big)$. Choose $\gamma$ and $h$ so that $\gamma \geqslant 5M^\lambda$ and $\sqrt{\kappa}\,\gamma h \leqslant 0.1$, where $\kappa = M^\lambda/m$. Assume that $\boldsymbol{\vartheta}_0^{\mathsf{KLMC}}$ is independent of $\mathbf{v}_0^{\mathsf{KLMC}}$ and that $\mathbf{v}_0^{\mathsf{KLMC}} \sim \mathcal{N}_p(0, \gamma\mathbf{I}_p)$. Then, for any $n \geqslant 1$, the distribution $\nu_n^{\mathsf{KLMC}}$ of $\boldsymbol{\vartheta}_n^{\mathsf{KLMC}}$ satisfies*

$$\mathsf{W}_2(\nu_n^{\mathsf{KLMC}}, \nu) \leqslant 2\varrho^n \mathsf{W}_2(\nu_0^{\mathsf{KLMC}}, \nu) + 3\sqrt{\frac{3\mu_2(\nu)\lambda(2p+2)}{r}} + 0.05\sqrt{\frac{\varrho^n \mathbb{E}[U^\lambda(\boldsymbol{\vartheta}_0^{\mathsf{KLMC}}) - f(\mathbf{0})]}{m}} + 0.9\gamma h\sqrt{\frac{\kappa p}{m}},$$

*where $\varrho = e^{-mh}$. Moreover, let the staring points $\boldsymbol{\vartheta}_0^{\mathsf{KLMC}} = \mathbf{0}$, it holds that*

$$\mathsf{W}_1(\nu_n^{\mathsf{KLMC}}, \nu) \leqslant 2\varrho^n \sqrt{\frac{p}{m}} + \frac{3\mu_1(\nu)\lambda(2p+1)}{r} + 0.9\gamma h\sqrt{\frac{\kappa p}{m}}.$$

To our knowledge, these results are the first reported convergence rate for the KLMC algorithm within the context of constrained sampling. Below, we set $M^\lambda = M + 1/\lambda^2$, aligning with the projection methods detailed in Example 2.1 and Example 2.2. We proceed by optimizing $\lambda$ to establish the subsequent corollary.

**Corollary 3.3.** *Let $\varepsilon \in (0, 1)$ be a small number. Set $\gamma = 5M^\lambda, M^\lambda = M + \frac{1}{\lambda^2}$.*

*(a) Set $\lambda = 3^{3/4}2^{1/8}r^{1/4}\big(\mu_1(\nu)(2p+1)\big)^{-1/4}p^{1/8}m^{-1/4}h^{1/4}$, choose $h > 0$ and $n \in \mathbb{N}$ so that*

$$h = 2^{-7/2}r^3\big(\mu_1(\nu)(2p+1)\big)^{-3}p^{3/2}m^{-1}\varepsilon^4 \quad and \quad n \geqslant \frac{1}{mh}\log(4/\varepsilon),$$

*then we have $\mathsf{W}_1(\nu_n^{\mathsf{KLMC}}, \nu) \leqslant \varepsilon\sqrt{p/m}$.*

*(b) Set $\lambda = 2^{1/7}3^{3/7}(r/\mu_2(\nu))^{1/7}(h/m)^{2/7}$, choose $h > 0$ and $n \in \mathbb{N}$ so that*

$$h = 32.26^{-7}m^{-5/2}r^3\mu_2(\nu)^{-3}\varepsilon^7 \quad and \quad n \geqslant \frac{1}{mh}\log(4/\varepsilon),$$

*then we have $\mathsf{W}_2(\nu_n^{\mathsf{KLMC}}, \nu) \leqslant \varepsilon\sqrt{p/m}$.*

We note that our error bounds depend in part on the synchronous coupling between the KLMC and the KLD 3.3. However, for the vanilla Langevin algorithm, Durmus et al. (2019) have shown that the dependency of the error bound on $\kappa$ can be improved by employing alternative couplings. We propose that similar improvements might be achievable for the KLMC algorithm in constrained sampling scenarios using non-synchronous coupling. For further insights, interested readers are directed to Yu et al. (2023).

### 3.4 Parallelized Randomized Midpoint Discretization of Kinetic Langevin Diffusion (pRKLMC)

The randomized midpoint method, introduced and studied in Shen & Lee (2019), aims at providing a discretization of the kinetic Langevin process 3.2 that reduces the bias of sampling as compared to more conventional discretizations. The parallel computing of this algorithm is outlined in Yu & Dalalyan (2024). This algorithm is referred to as pRKLMC, and for the convenience of the readers, we restate it in Algorithm 2. To ease the notation, we omit the superscript pRKLMC therein. In

---

**Algorithm 2** Parallelized RKLMC (pRKLMC)

---

**Input**: number of parallel steps $R$, number of sequential iterations $Q$, step size $h$, friction coefficient $\gamma$, number of iterations $K$, initial points $\mathbf{v}_0$ and $\boldsymbol{\vartheta}_0$.

**Output**: iterates $\boldsymbol{\vartheta}_{K+1}$ and $\mathbf{v}_{K+1}$

1: **for** $k = 1$ to $K$ **do**
2: $\quad$ Draw $U_{k1}, \ldots, U_{kR}$ uniformly from $\left[0, \frac{1}{R}\right], \ldots, \left[\frac{R-1}{R}, 1\right]$, respectively.
3: $\quad$ Generate $\bar{\mathbf{W}}_s = \mathbf{W}_{kh+s} - \mathbf{W}_{kh}$
4: $\quad$ Generate $\boldsymbol{\xi}_{kr} = \int_0^{U_{kr}h} \left(1 - e^{-\gamma(U_{kr}h-s)}\right)d\bar{\mathbf{W}}_s, r = 1, \ldots, R.$
5: $\quad$ Set $\boldsymbol{\vartheta}_k^{(0,r)} = \boldsymbol{\vartheta}_k, r = 1, \ldots, R.$
6: $\quad$ **for** $q = 1$ to $Q - 1$ **do**
7: $\quad\quad$ **for** $r = 1$ to $R$ in parallel **do**
8: $\quad\quad\quad a_{kr} = \frac{1-e^{-\gamma h U_{kr}}}{\gamma}.$
9: $\quad\quad\quad b_{kj} = \int_{\frac{(j-1)h}{R}}^{h\min(\frac{j}{R}, U_{kr})} (1 - e^{-\gamma(U_{kr}h-s)})ds, j = 1, \ldots, r.$
10: $\quad\quad\quad \boldsymbol{\vartheta}_k^{(q,r)} = \boldsymbol{\vartheta}_k + a_{kr}\mathbf{v}_k - \sum_{j=1}^r b_{kj}\nabla U^\lambda(\boldsymbol{\vartheta}_k^{(q-1,j)}) + \sqrt{2}\boldsymbol{\xi}_{kr}.$
11: $\quad\quad$ **end for**
12: $\quad$ **end for**
13: $\quad \boldsymbol{\vartheta}_{k+1} = \boldsymbol{\vartheta}_k + \frac{1-e^{-\gamma h}}{\gamma}\mathbf{v}_k - \sum_{r=1}^R \frac{h}{R}(1 - e^{-\gamma h(1-U_{kr})})\nabla U^\lambda(\boldsymbol{\vartheta}_k^{(Q-1,r)}) + \sqrt{2}\int_0^h (1 - e^{-\gamma(h-s)})d\bar{\mathbf{W}}_s.$
14: $\quad \mathbf{v}_{k+1} = e^{-\gamma h}\mathbf{v}_k - \gamma\sum_{r=1}^R \frac{h}{R}e^{-\gamma h(1-U_{kr})}\nabla U^\lambda(\boldsymbol{\vartheta}_k^{(Q-1,r)}) + \sqrt{2}\gamma\int_0^h e^{-\gamma(h-s)}d\bar{\mathbf{W}}_s.$
15: **end for**

(In parallel) — lines 8–10

---

the theorem below, we quantify the error bounds for this algorithm when applied to constrained sampling.

**Theorem 3.4.** *Under Assumptions 2.1 and 2.2, we further assume that the potential $U^\lambda$ satisfies Condition 2.1. Let the tuning parameter $\lambda \in \left(0, \frac{r}{2(p+2)}\right)$. Choose $\gamma$ and $h$ so that $\gamma \geqslant 5M^\lambda$ and $\gamma h \leqslant 0.1\kappa^{-1/6}$, where $\kappa = M^\lambda/m$. Assume that $\boldsymbol{\vartheta}_0^{\mathrm{pRKLMC}}$ is independent of $\mathbf{v}_0^{\mathrm{pRKLMC}}$ and that $\mathbf{v}_0^{\mathrm{pRKLMC}} \sim \mathcal{N}_p(0, \gamma\mathbf{I}_p)$. Then, for any $n \geqslant 1$, the distribution $\nu_n^{\mathrm{pRKLMC}}$ of $\boldsymbol{\vartheta}_n^{\mathrm{pRKLMC}}$ satisfies*

$$W_2(\nu_n^{\mathrm{pRKLMC}}, \nu) \leqslant 1.8\varrho^n W_2(\nu_0^{\mathrm{pRKLMC}}, \nu) + 2.8\sqrt{\frac{3\mu_2(\nu)\lambda(2p+2)}{r}} + 0.28\sqrt{\frac{\varrho^n \mathbb{E}[U^\lambda(\boldsymbol{\vartheta}_0^{\mathrm{pRKLMC}}) - f(\mathbf{0})]}{m}}$$

$$+ 44.78\sqrt{\frac{(\gamma M^\lambda h^2)^3}{R^2} + (\gamma M^\lambda h^2)^{2Q-1}}\sqrt{\frac{\kappa p}{m}},$$

*where $\varrho = e^{-mh}$. Moreover, let the staring points $\boldsymbol{\vartheta}_0^{\mathrm{pRKLMC}} = \mathbf{0}$, it holds that*

$$W_1(\nu_n^{\mathrm{pRKLMC}}, \nu) \leqslant 1.8\varrho^n\sqrt{\frac{p}{m}} + \frac{3\mu_1(\nu)\lambda(2p+1)}{r} + 44.78\sqrt{\frac{(\gamma M^\lambda h^2)^3}{R^2} + (\gamma M^\lambda h^2)^{2Q-1}}\sqrt{\frac{\kappa p}{m}}.$$

To our knowledge, this represents the first convergence analysis for the pRKLMC algorithm within the context of constrained sampling. When $R = 1$, we recover the convergence rate for RKLMC Shen & Lee (2019); Yu et al. (2023); Yu & Dalalyan (2024) when applied in the context of constrained sampling. We define $M^\lambda = M + 1/\lambda^2$, in line with the projection options described in Examples 2.1 and 2.2. We then select the optimal $\lambda$ to derive the following corollary.

**Corollary 3.4.** *Given the number of parallel steps $R$, set $Q = \lceil \log R \rceil + 2$. Let $\varepsilon \in (0,1)$ be a small number, $\gamma = 5M^\lambda$ with $M^\lambda = M + 1/\lambda^2$.*

*(a) Set $\lambda = \left(\frac{7 \cdot 8011}{3}\right)^{1/8}\left(\frac{rh^3}{R\mu_1(\nu)(2p+1)}\right)^{1/8}\left(\frac{p}{m^2}\right)^{1/16}$, choose $h > 0$ and $n \in \mathbb{N}$ so that*

$$h = 23.6^{-8/3}p^{7/6}m^{-1}r^{7/3}\big(\mu_1(\nu)(2p+1)\big)^{-7/3}R^{1/3}\varepsilon^{8/3} \quad \text{and} \quad n \geqslant \frac{1}{mh}\log\left(\frac{3.6}{\varepsilon}\right),$$

*then we have* $\mathsf{W}_1(\nu_n^{\mathsf{pRKLMC}}, \nu) \leqslant \varepsilon\sqrt{p/m}$.

**(b)** *Set* $\lambda = (\frac{14 \cdot 8011}{5})^{2/15}(\frac{rp}{m^2\mu_2(\nu)(2p+2)})^{1/15}(\frac{h^3}{R})^{2/15}$, *choose* $h > 0$ *and* $n \in \mathbb{N}$ *so that*

$$h = 20.9^{-5}p^{7/3}m^{-13/6}r^{7/3}\big(\mu_2(\nu)(2p+2)\big)^{-7/3}R^{1/3}\varepsilon^5 \quad and \quad n \geqslant \frac{1}{mh}\log\left(\frac{3.6}{\varepsilon}\right),$$

*then we have* $\mathsf{W}_2(\nu_n^{\mathsf{pRKLMC}}, \nu) \leqslant \varepsilon\sqrt{p/m}$.

The corollary demonstrates that pRKLMC achieves the best performance in terms of sample complexity compared to the other three algorithms considered in this work.

## 4 DISCUSSION

**Strongly convexity**    In this work, we focus on a strongly convex potential function $f$ coupled with specific choices of projection operators, resulting in a strongly log-concave surrogate density $\nu^\lambda$. However, it is important to note that the approximation error bound between the target density $\nu$ and the surrogate density $\nu^\lambda$, as stated in Proposition 2.1, depends solely on the convexity of $f$. By relaxing the strong convexity requirement of $f$ and considering alternative projection operators, one can generate a surrogate density $\nu^\lambda$ that is less restrictive than strongly log-concave. This modification enables the extension of these findings to a broader spectrum of sampling methods.

**Other metrics**    The Wasserstein distance we employ is a natural metric for measuring sampling errors due to its relevance to optimal transport theory. However, recent advancements in gradient-based sampling have investigated other metrics, including total variation distance, KL divergence, and $\chi^2$ divergence. An intriguing avenue for future research would be to establish error guarantees for constrained sampling concerning these alternative metrics.

**Limitation**    The primary focus of this work is to provide theoretical insights into the analysis of constrained sampling. We hope that these insights pave the way for empirical evaluations of the performance of Langevin-type algorithms in various settings, as well as for implementations in real data applications, which offer promising directions for future research.

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
