## A    Relation to Previous Works on Constrained Sampling

Sampling from constrained log-concave distributions is a fundamental task across various fields. In computer science, several studies have explored the convergence of the ball walk and hit-and-run algorithms toward the uniform density on a convex body $\mathcal{K}$, or more generally, a log-concave density (Dyer et al., 1991; Kannan et al., 1997; Kook et al., 2024; Lovász, 1999; Lovász & Simonovits, 1993; Lovász & Vempala, 2007; Smith, 1984). Unlike the first-order sampling methods examined in this work, these algorithms require calls to zero-order oracles at each iteration. Additionally, the performance of these geometric random walks is influenced by the skewed shape of $\mathcal{K}$, necessitating pre-processing steps to enhance efficiency.

Another category of samplers addresses the geometric structure of convex constraints, including Riemannian Hamiltonian Monte Carlo (Brubaker et al., 2012; Gürbüzbalaban et al., 2022; Kook et al., 2022), Gibbs sampling (Gelfand et al., 1992), and Mirror Langevin methods (Ahn & Chewi, 2021; Chewi et al., 2020; Hsieh et al., 2018; Zhang et al., 2020).

A particularly relevant approach to this work involves diffusion-type samplers (Brosse et al., 2017; Bubeck et al., 2015; 2018; Lamperski, 2021; Lehec, 2021; 2023). These samplers, based on discretizations of Itô diffusions, exhibit rapid convergence to the target density in continuous time and have been extensively employed in unconstrained settings. While the underlying stochastic processes can be generalized to constrained settings, the discretization analysis heavily relies on the smoothness of the target distribution, which is challenging to achieve in the constrained context considered here.

Among these works, Brosse et al. (2017) is the most closely aligned with the spirit of this work. It considers the same setting as the current study, proposing the use of the Moreau envelope (Example 2.1) to derive a smooth surrogate density, followed by applying LMC to sample from this density. Their approach supports our general framework, which integrates Euclidean projection with LMC, and provides convergence bounds in both the $W_1$ metric and total variation norm. Our work expands on this approach by presenting a general scheme that accommodates a broader range of projection methods and sampling techniques beyond LMC. This enhances the applicability of algorithms such as KLMC, pRLMC, and pRKLMC in constrained settings. Furthermore, we establish a lower bound for the distance between the surrogate distribution and the target distribution in Proposition 2.2, demonstrating the tightness of the bound for the distance between the approximation and the target density established in Proposition 2.1. This aspect has not been addressed in any of the previously mentioned works on the diffusion-type samplers.

## B    Proofs of Section 2

In this part, we provide the proof of Lemma 2.1, Lemma 2.2, Proposition 2.1, and Proposition 2.2.

### B.1    Proofs of Lemma 2.1 and Lemma 2.2

*Proof of Lemma 2.1.* By Theorem 2.26 in Rockafellar & Wets (2009), the function $\ell_{\mathcal{K}}^{E,\lambda}(x)$ is convex and continuously differentiable, and $\ell_{\mathcal{K}}^{E,\lambda}(x)$ is $\frac{1}{\lambda^2}$-smooth. The last claim follows from Lemma 1 in Brosse et al. (2017).

□

*Proof of Lemma 2.2.* By the definition of $\ell_{\mathcal{K}}^{G,\lambda}$, we have

$$\ell_{\mathcal{K}}^{G,\lambda} = \frac{1}{2\lambda^2}\Big(1 - \frac{1}{g_{\mathcal{K}}(x)}\Big)^2\|x\|_2^2.$$

By the definition of $U^{G,\lambda}$, the function $U^{G,\lambda}$ is $m$-strongly convex, continuously differentiable. Moreover, it holds that

$$\|\nabla\ell_{\mathcal{K}}^{G,\lambda}(x) - \nabla\ell_{\mathcal{K}}^{G,\lambda}(y)\|_2 \leqslant \frac{1}{\lambda^2}\|x - y\|_2,$$

This implies $U^{G,\lambda}$ is $\frac{1}{\lambda^2} + M$-smooth. Now we show that $\int_{\mathbb{R}^p} e^{-U^{G,\lambda}(s)} \mathrm{d}s < \infty$. Let $R :=$ $\max_{x \in \mathcal{K}} \|x\|_2$, and there is a constant $l \in \mathbb{R}$ such that $f(x) \geqslant l, \forall x \in \mathbb{R}$. Define $R := \sup_{x \in \mathcal{K}} \|x\|_2$. Note that

$$\int_{\mathbb{R}^p} e^{-U^{G,\lambda}(x)} \mathrm{d}x = \int_{\mathcal{K}} e^{-f(x)} \mathrm{d}x + \int_{\mathcal{B}_2(0,R) \cap \mathcal{K}^c} e^{-U^{G,\lambda}(x)} \mathrm{d}x + \int_{\mathcal{B}_2(0,R)^c} e^{-U^{G,\lambda}(x)} \mathrm{d}x \,.$$

We now derive the bound for the second and the third terms on the right-hand side of the previous display. Since $f$ is convex, there exists $a \in \mathbb{R}$ and $b \in \mathbb{R}^p$ such that

$$f(x) \geqslant a + \langle b, x \rangle \,.$$

We then find

$$\int_{\mathcal{B}_2(0,R) \cap \mathcal{K}^c} e^{-U^{G,\lambda}(x)} \mathrm{d}x \leqslant e^{-a+bR} \int_{\mathcal{B}_2(0,R)} \mathrm{d}x \,.$$

Thus,

$$\begin{aligned}
\int_{\mathcal{B}_2(0,R) \cap \mathcal{K}^c} e^{-U^{G,\lambda}(x)} \mathrm{d}x &\leqslant e^{-a+bR} \int_{\mathcal{B}_2(0,R)} \mathrm{d}x \\
&\leqslant e^{-a+bR} \int_{\mathcal{B}_2(0,R)} \mathrm{d}x \\
&= e^{-a+bR} \frac{\pi^{p/2}}{\Gamma(\frac{d}{2}+1)} R^p \\
&< \infty \,.
\end{aligned}$$

Note that

$$\int_{\mathcal{B}_2(0,R)^c} e^{-U^{G,\lambda}(x)} \mathrm{d}x \leqslant e^{-l} \int_{\mathcal{B}_2(0,R)^c} e^{-\frac{1}{2\lambda^2}(\|x\|_2 - R)^2} \mathrm{d}x$$

By Fubini's theorem and the fact that

$$e^{-\frac{1}{2\lambda^2}(\|x\|_2 - R)^2} = \int_0^\infty \frac{t}{\lambda^2} e^{-\frac{t^2}{2\lambda^2}} \mathbf{1}_{[\|x\|_2 - R, \infty)}(t) \mathrm{d}t \,,$$

we obtain

$$\begin{aligned}
\int_{\mathcal{B}_2(0,R)^c} e^{-U^{G,\lambda}(x)} \mathrm{d}x &\leqslant e^{-l} \int_0^\infty \frac{t}{\lambda^2} e^{-\frac{t}{2\lambda^2}} \int_{x:\|x\|_2 \geq R} \mathbf{1}_{[\|x\|_2 - R, \infty)}(t) \mathrm{d}x \mathrm{d}t \\
&\leqslant e^{-l} \int_0^\infty \frac{t}{\lambda^2} e^{-\frac{t}{2\lambda^2}} \int_{\mathcal{B}_2(0,R+t)} \mathrm{d}x \mathrm{d}t \\
&= e^{-l} \int_0^\infty \frac{t}{\lambda^2} e^{-\frac{t}{2\lambda^2}} \frac{\pi^{p/2}}{\Gamma(\frac{p}{2}+1)} (R+t)^d \mathrm{d}t \\
&< \infty \,.
\end{aligned}$$

Collecting pieces gives the desired result.

$\square$

## B.2 PROOFS OF PROPOSITION 2.1

To prove the Proposition 2.1, we need the following auxiliary lemma.

**Lemma B.1.** *Assume $f$ is convex and $\mathbf{0} = \arg\min_{\boldsymbol{\theta} \in \mathcal{K}} f(\boldsymbol{\theta})$. Under Assumption 2.1 , it holds for any $\lambda \in \left(0, \frac{r}{2(p+k)}\right)$ that*

$$\int_{\mathcal{K}^c} \|x\|_2^k \nu^\lambda(x) \mathrm{d}x \leqslant \frac{3\lambda(p+k)}{r} \times \mu_k(\nu) \,,$$

*where $\mu_k(\nu) = \int_{\mathbb{R}^p} \|\boldsymbol{\theta}\|_2^k \nu(\mathrm{d}\boldsymbol{\theta})$.*

*Proof of Lemma B.1.* By Fubini's theorem and the fact that

$$\int_0^\infty \frac{t}{\lambda^2} e^{-\frac{t^2}{2\lambda^2}} \mathbf{1}_{[\|x-\mathsf{P}_\mathcal{K}(x)\|_2,+\infty)}(t)\mathrm{d}t = e^{-\frac{1}{2\lambda^2}\|x-\mathsf{P}_\mathcal{K}(x)\|_2^2},$$

it holds for every $k \geqslant 0$ that

$$\int_{\mathcal{K}^c} \|x\|_2^k e^{-U^\lambda(x)}\mathrm{d}x = \int_0^\infty \Big( \underbrace{\int_{\mathcal{K}^c} \|x\|_2^k e^{-f(x)} \mathbf{1}(\|x - \mathsf{P}_\mathcal{K}(x)\|_2 \leqslant t)\,\mathrm{d}x}_{\psi(t)} \Big) \frac{t}{\lambda^2} e^{-\frac{t^2}{2\lambda^2}}\,\mathrm{d}t. \qquad \text{(B.1)}$$

Define the set $\mathcal{K}_t := \{x \in \mathbb{R}^p : \|x - \mathsf{P}_\mathcal{K}(x)\|_2 \leqslant t\}$, we then have

$$\psi(t) = \int_{\mathcal{K}_t} \|x\|_2^k e^{-f(x)}\,\mathrm{d}x - \int_\mathcal{K} \|x\|_2^k e^{-f(x)}\,\mathrm{d}x. \qquad \text{(B.2)}$$

On the other hand, it holds for every $b > 0$ that

$$\int_{\mathcal{K}_t} \|x\|_2^k e^{-f(x)}\,\mathrm{d}x = (1+b)^{p+k} \int_{\mathcal{K}_t/(1+b)} \|y\|_2^k e^{-f((1+b)y)}\,\mathrm{d}y$$

$$\leqslant (1+b)^{p+k} \int_{\mathcal{K}_t/(1+b)} \|y\|_2^k e^{-f(y)}\,\mathrm{d}y$$

where we have used the inequality

$$f((1+b)y) + bf(0) \geqslant (1+b)f\Big(\frac{1}{1+b}(1+b)y + \frac{b}{b+1}0\Big) = (1+b)f(y) \geqslant f(y) + bf(0),$$

implying that $f((1+b)y) \geqslant f(y)$ for every $b > 0$ and every $y \in \mathbb{R}^p$. In addition, it holds for any $x \in \mathcal{K}_t/(1+b)$ that

$$\|(1+b)x - \mathsf{P}_\mathcal{K}\big((1+b)x\big)\|_2 \leqslant t$$

which is equivalent to

$$\frac{1+b}{b}\|x - \frac{1}{1+b}\mathsf{P}_\mathcal{K}\big((1+b)x\big)\|_2 \leqslant \frac{t}{b}.$$

Note that we can rewrite $x$ as

$$x = \frac{1}{1+b}\mathsf{P}_\mathcal{K}\big((1+b)x\big) + \frac{b}{1+b}\frac{x - \frac{1}{1+b}\mathsf{P}_\mathcal{K}\big((1+b)x\big)}{\frac{b}{1+b}}.$$

Notice that

$$\mathsf{P}_\mathcal{K}\big((1+b)x\big) \in \mathcal{K}$$

and when $b = t/r$ we have

$$\frac{x - \frac{1}{1+b}\mathsf{P}_\mathcal{K}\big((1+b)x\big)}{\frac{b}{1+b}} \in \mathcal{K}.$$

By the convexity of the set $\mathcal{K}$, this implies $x \in \mathcal{K}$. Therefore, we obtain

$$\int_{\mathcal{K}_t} \|x\|_2^k e^{-f(x)}\,\mathrm{d}x \leqslant \Big(1 + \frac{t}{r}\Big)^{p+k} \int_\mathcal{K} \|y\|_2^k e^{-f(y)}\,\mathrm{d}y \leqslant e^{(p+k)t/r} \int_\mathcal{K} \|y\|_2^k e^{-f(y)}\,\mathrm{d}y.$$

Combining this inequality with display B.2, we get

$$\psi(t) \leqslant \big(e^{(p+k)t/r} - 1\big) \int_\mathcal{K} \|y\|_2^k e^{-f(y)}\,\mathrm{d}y.$$

Substituting this upper bound in display B.1 leads to

$$
\begin{aligned}
\int_{\mathcal{K}^c} \|x\|_2^k e^{-U^\lambda(x)} \mathrm{d}x
&\leqslant \int_0^\infty \frac{t}{\lambda^2} e^{-t^2/2\lambda^2} \left( e^{(p+k)t/r} - 1 \right) \mathrm{d}t \times \int_{\mathcal{K}} \|y\|_2^k e^{-f(y)} \,\mathrm{d}y \\
&= \int_0^\infty \left( e^{(p+k)t/r} - 1 \right) \mathrm{d}\left( -e^{-t^2/2\lambda^2} \right) \times \int_{\mathcal{K}} \|y\|_2^k e^{-f(y)} \,\mathrm{d}y \\
&= \frac{(p+k)}{r} \int_0^\infty \exp\left\{ -\frac{t^2}{2\lambda^2} + \frac{(p+k)t}{r} \right\} \mathrm{d}t \times \int_{\mathcal{K}} \|y\|_2^k e^{-f(y)} \,\mathrm{d}y \\
&= \frac{(p+k)}{r} \exp\left\{ \frac{\lambda^2(p+k)^2}{2r^2} \right\} \int_0^\infty \exp\left\{ -\frac{(t - (\lambda^2/r)(p+k))^2}{2\lambda^2} \right\} \mathrm{d}t \times \int_{\mathcal{K}} \|y\|_2^k e^{-f(y)} \,\mathrm{d}y \\
&\leqslant \frac{\sqrt{2\pi}\lambda\,(p+k)}{r} \exp\left\{ \frac{\lambda^2(p+k)^2}{2r^2} \right\} \times \int_{\mathcal{K}} \|y\|_2^k e^{-f(y)} \,\mathrm{d}y \,.
\end{aligned}
$$

When $\lambda < \frac{r}{2(p+k)}$, it holds that

$$
\int_{\mathcal{K}^c} \|x\|_2^k e^{-U^\lambda(x)} \mathrm{d}x \leqslant \frac{3\lambda\,(p+k)}{r} \times \int_{\mathcal{K}} \|y\|_2^k e^{-f(y)} \,\mathrm{d}y. \tag{B.3}
$$

Finally, note that the normalizing constant of $\nu^\lambda$ can be lower bounded as follows

$$
\int_{\mathbb{R}^p} e^{-U^\lambda(x)} \mathrm{d}x \geqslant \int_{\mathcal{K}} e^{-U^\lambda(x)} \mathrm{d}x = \int_{\mathcal{K}} e^{-f(x)} \mathrm{d}x \,.
$$

The desired result follows readily by dividing the two sides of display B.3 by $\int_{\mathbb{R}^p} e^{-U^\lambda(x)} \mathrm{d}x$ and employing this inequality.

$\square$

We are now ready to prove Proposition 2.1

*Proof of Proposition 2.1.* By Theorem 6.15 in Villani (2009), it holds that

$$
\mathsf{W}_q^q(\nu, \nu^\lambda) \leqslant \int_{\mathbb{R}^p} \|x\|_2^q |\nu(x) - \nu^\lambda(x)| \mathrm{d}x \,.
$$

We note that

$$
\begin{aligned}
\int_{\mathbb{R}^p} \|x\|_2^q |\nu(x) - \nu^\lambda(x)| \,\mathrm{d}x
&= \int_{\mathcal{K}} \|x\|_2^q |\nu(x) - \nu^\lambda(x)| \mathrm{d}x + \int_{\mathcal{K}^c} \|x\|_2^q \nu^\lambda(x) \mathrm{d}x \\
&= \left( 1 - \frac{\int_{\mathcal{K}} e^{-f(y)} \,\mathrm{d}y}{\int_{\mathbb{R}^p} e^{-U^\lambda(y)} \,\mathrm{d}y} \right) \int_{\mathcal{K}} \|x\|_2^q \nu(x) \mathrm{d}x + \int_{\mathcal{K}^c} \|x\|_2^q \nu^\lambda(x) \mathrm{d}x \\
&= \frac{\int_{\mathcal{K}^c} e^{-U^\lambda(y)} \,\mathrm{d}y}{\int_{\mathbb{R}^p} e^{-U^\lambda(y)} \,\mathrm{d}y} \times \int_{\mathcal{K}} \|x\|_2^q \nu(x) \mathrm{d}x + \int_{\mathcal{K}^c} \|x\|_2^q \nu^\lambda(x) \mathrm{d}x \\
&= \int_{\mathcal{K}^c} \nu^\lambda(y) \,\mathrm{d}y \times \mu_q(\nu) + \int_{\mathcal{K}^c} \|x\|_2^q \nu^\lambda(x) \mathrm{d}x \,.
\end{aligned}
$$

When $\lambda \leqslant \frac{r}{2(p+q)}$, we apply Lemma B.1 with $k = 0$ and $k = q$, which leads us to the inequality

$$
\int_{\mathbb{R}^p} \|x\|_2^q |\nu(x) - \nu^\lambda(x)| \,\mathrm{d}x \leqslant \frac{3\lambda\big(p\mu_0(\nu) + (p+q)\big)\mu_q(\nu)}{r} = \frac{3\lambda(2p+q)}{r} \times \mu_q(\nu).
$$

This completes the proof.

$\square$

### B.3 PROOF OF PROPOSITION 2.2

To prove Proposition 2.2, we need the following auxiliary lemma.

**Lemma B.2.** *Under the assumptions stated in Proposition 2.2, for any $k \geqslant 1$ and any $\lambda \in \left(0, \frac{r}{2(p+k)} \wedge \frac{1}{\sqrt{M}}\right)$, it holds that*

$$\int_{\mathcal{K}^c} \|x\|_2^k \nu^\lambda(\mathrm{d}x)/\mu^k(\nu) \geqslant \frac{1}{2}\sqrt{\frac{\pi}{2}} e^{-\frac{3}{4}Mr^2} \frac{p+k}{r+3\lambda p}\lambda \, .$$

*Proof of Lemma B.2.* By the definition of $\nu^\lambda$ and $\mu^k(\nu)$, we have

$$\int_{\mathcal{K}^c} \|x\|_2^k \nu^\lambda(\mathrm{d}x)/\mu^k(\nu) = \frac{\int_{\mathcal{K}^c} \|x\|_2^k e^{-f(x)-\frac{1}{2\lambda^2}(\|x\|_2-r)^2}\mathrm{d}x}{\int_{\mathbb{R}^p} e^{-U^\lambda(x)}\mathrm{d}x} \cdot \frac{\int_{\mathcal{K}} e^{-f(x)}\mathrm{d}x}{\int_{\mathcal{K}} \|x\|_2^k e^{-f(x)}\mathrm{d}x} \, . \tag{B.4}$$

Note that $f$ is $M$-smooth and $f(\mathbf{0}) = 0$, it then holds that

$$f(x) \leqslant \frac{M}{2}\|x\|_2^2 \, .$$

This implies

$$\int_{\mathcal{K}^c} \|x\|_2^k e^{-f(x)-\frac{1}{2\lambda^2}(\|x\|_2-r)^2}\mathrm{d}x \geqslant \int_{\mathcal{K}^c} \|x\|_2^k e^{-\frac{M}{2}\|x\|_2^2-\frac{1}{2\lambda^2}(\|x\|_2-r)^2}\mathrm{d}x \, .$$

Employing the integration in polar coordinates gives

$$\int_{\mathcal{K}^c} \|x\|_2^k e^{-\frac{M}{2}\|x\|_2^2-\frac{1}{2\lambda^2}(\|x\|_2-r)^2}\mathrm{d}x = \omega_{p-1}\int_r^{+\infty} t^{k+p-1}e^{-\frac{M}{2}t^2-\frac{1}{2\lambda^2}(t-r)^2}\mathrm{d}t \, ,$$

where $\omega_{p-1} = \int_{-\pi}^{\pi}\int_0^{\pi}\cdots\int_0^{\pi}\prod_{j=1}^{p-2}(\sin\theta_j)^{p-j-1}\mathrm{d}\theta_1\cdots\mathrm{d}\theta_{p-1}$. Let $t = r + \lambda z$, it then follows that

$$\int_{\mathcal{K}^c} \|x\|_2^k e^{-f(x)-\frac{1}{2\lambda^2}(\|x\|_2-r)^2}\mathrm{d}x$$

$$\geqslant \lambda\omega_{p-1}\int_0^{+\infty} (r+\lambda z)^{k+p-1}e^{-\frac{M}{2}(r+\lambda z)^2-\frac{1}{2}z^2}\mathrm{d}z$$

$$\geqslant \lambda\omega_{p-1}r^{k+p-1}\int_0^{+\infty} e^{-\frac{M}{2}(r+\lambda z)^2-\frac{1}{2}z^2}\mathrm{d}z$$

$$= \lambda\omega_{p-1}r^{k+p-1}\int_0^{+\infty} e^{-\left(\sqrt{\frac{1+M\lambda^2}{2}}x+\frac{M\lambda r}{\sqrt{2(1+M\lambda^2)}}\right)^2}\mathrm{d}x e^{-\frac{Mr^2}{2}\left(1+\frac{M\lambda^2}{1+M\lambda^2}\right)} \, .$$

The last step follows from a change of variables. Using the fact that $\lambda < 1/\sqrt{M}$ and $(x+y)^2 \leqslant 2x^2 + 2y^2, \forall x, y \geqslant 0$, we then obtain

$$\int_{\mathcal{K}^c} \|x\|_2^k e^{-f(x)-\frac{1}{2\lambda^2}(\|x\|_2^2-r)^2}\mathrm{d}x \geqslant \lambda\omega_{p-1}r^{k+p-1}\int_0^{+\infty} e^{-(1+M\lambda^2)x^2}\mathrm{d}x e^{-\frac{Mr^2}{2}(1+\frac{M\lambda^2}{1+M\lambda^2})}$$

$$= \lambda\omega_{p-1}r^{k+p-1}\frac{1}{2}\sqrt{\frac{\pi}{1+M\lambda^2}}e^{-\frac{Mr^2}{2}(1+\frac{M\lambda^2}{1+M\lambda^2})}$$

$$\geqslant \lambda\omega_{p-1}r^{k+p-1}\frac{1}{2}\sqrt{\frac{\pi}{2}}e^{-\frac{3}{4}Mr^2} \, .$$

Moreover, by display B.3, we find

$$\int_{\mathbb{R}^p} e^{-U^\lambda(x)}\mathrm{d}x = \int_{\mathcal{K}} e^{-U^\lambda(x)}\mathrm{d}x + \int_{\mathcal{K}^c} e^{-U^\lambda(x)}\mathrm{d}x$$

$$\leqslant \left(1 + \frac{3\lambda p}{r}\right)\int_{\mathcal{K}} e^{-\|x\|_2^2/2}\mathrm{d}x \, .$$

Furthermore, using the integration in polar coordinates again gives

$$\int_{\mathcal{K}} \|x\|_2^k e^{-f(x)} \mathrm{d}x \leqslant \omega_{p-1} \int_0^r t^{k+p-1} \mathrm{d}t$$
$$= \omega_{p-1} \frac{r^{k+p}}{k+p}.$$

Collecting pieces and plugging them into display B.4 gives

$$\int_{\mathcal{K}^c} \|x\|_2^k \nu^\lambda(\mathrm{d}x)/\mu^k(\nu) \geqslant \frac{1}{2}\sqrt{\frac{\pi}{2}} e^{-\frac{3}{4}Mr^2} \frac{p+k}{r+3\lambda p}\lambda$$

as desired.

$\square$

We are now ready to prove Proposition 2.2.

*Proof of Proposition 2.2.* By Proposition 7.29 in Villani (2009), it holds for any $q \geqslant 1$ that

$$\mathsf{W}_q^q(\nu^\lambda, \nu) \geqslant \left| \left( \int_{\mathbb{R}^p} \|x\|_2^q \nu^\lambda(\mathrm{d}x) \right)^{1/q} - \left( \int_{\mathbb{R}^p} \|x\|_2^q \nu(\mathrm{d}x) \right)^{1/q} \right|^q.$$

Dividing both sides by $\mu_q(\nu)$ gives

$$\mathsf{W}_q^q(\nu^\lambda, \nu)/\mu_q(\nu) \geqslant \left| \left( \int_{\mathbb{R}^p} \|x\|_2^q \nu^\lambda(\mathrm{d}x)/\mu_q(\nu) \right)^{1/q} - 1 \right|^q$$
$$= \left| \left( \frac{\int_{\mathcal{K}} \|x\|_2^q \nu^\lambda(\mathrm{d}x) + \int_{\mathcal{K}^c} \|x\|_2^q \nu^\lambda(\mathrm{d}x)}{\int_{\mathbb{R}^p} \|x\|_2^q \nu(\mathrm{d}x)} \right)^{1/q} - 1 \right|.$$

By the definitions of $\nu$ and $\nu^\lambda$, we have

$$\frac{\int_{\mathcal{K}} \|x\|_2^q \nu^\lambda(\mathrm{d}x)}{\int_{\mathbb{R}^p} \|x\|_2^q \nu(dx)} = \frac{\int_{\mathcal{K}} \|x\|_2^q e^{-f(x)} \mathrm{d}x}{\int_{\mathbb{R}^p} e^{-U^\lambda} dx} \cdot \frac{\int_{\mathcal{K}} e^{-f(x)} \mathrm{d}x}{\int_{\mathcal{K}} \|x\|_2^q e^{-f(x)} \mathrm{d}x}$$
$$= \frac{\int_{\mathcal{K}} e^{-f(x)} \mathrm{d}x}{\int_{\mathcal{K}} e^{-f(x)} \mathrm{d}x + \int_{\mathcal{K}^c} e^{-f(x)} \mathrm{d}x}.$$

Combining this with display B.3 then yields

$$\frac{\int_{\mathcal{K}} \|x\|_2^q \nu^\lambda(\mathrm{d}x)}{\int_{\mathbb{R}^p} \|x\|_2^q \nu(\mathrm{d}x)} \geqslant \frac{r}{r+3\lambda p}.$$

By Lemma B.2 and Lemma B.1, we have

$$\frac{1}{2}\sqrt{\frac{\pi}{2}} e^{-\frac{3}{4}Mr^2} \frac{p+q}{r+3\lambda p}\lambda \leqslant \frac{\int_{\mathcal{K}^c} \|x\|_2^q \nu^\lambda(\mathrm{d}x)}{\int_{\mathbb{R}^p} \|x\|_2^q \nu(\mathrm{d}x)} \leqslant \frac{3\lambda(p+q)}{r}.$$

Collecting pieces gives the desired result.

$\square$

## C  PROOFS OF SECTION 3

*Proof of Theorem 3.1.* The triangle inequality for the Wasserstein distance provides us with

$$\mathsf{W}_2(\nu_n^{\mathsf{LMC}}, \nu) \leqslant \mathsf{W}_2(\nu_n^{\mathsf{LMC}}, \nu^\lambda) + \mathsf{W}_2(\nu^\lambda, \nu). \tag{C.1}$$

By Theorem 9 in Durmus et al. (2019), it holds that

$$\mathsf{W}_2(\nu_n^{\mathsf{LMC}}, \nu^\lambda) \leqslant e^{-\frac{mnh}{2}} \mathsf{W}_2(\nu_0^{\mathsf{LMC}}, \nu^\lambda) + \sqrt{\frac{2M^\lambda ph}{m}} \tag{C.2}$$

provided that $h \leqslant 1/M^{\lambda}$. Using the triangle inequality for the Wasserstein distance again gives

$$\mathsf{W}_2(\nu_n^{\mathsf{LMC}}, \nu^{\lambda}) \leqslant e^{-\frac{mnh}{2}}\mathsf{W}_2(\nu_0^{\mathsf{LMC}}, \nu) + \sqrt{\frac{2M^{\lambda}ph}{m}} + \mathsf{W}_2(\nu, \nu^{\lambda}).$$

Plugging this back to display C.1 then provides us with

$$\mathsf{W}_2(\nu_n^{\mathsf{LMC}}, \nu) \leqslant e^{-\frac{mnh}{2}}\mathsf{W}_2(\nu_0^{\mathsf{LMC}}, \nu) + \sqrt{\frac{2M^{\lambda}ph}{m}} + 2\mathsf{W}_2(\nu, \nu^{\lambda}).$$

Combining this with Proposition 2.1, for any $\lambda \in \left(0, \frac{r}{2(p+2)}\right)$, we obtain

$$\mathsf{W}_2(\nu_n^{\mathsf{LMC}}, \nu) \leqslant e^{-\frac{mnh}{2}}\mathsf{W}_2(\nu_0^{\mathsf{LMC}}, \nu) + \sqrt{\frac{2M^{\lambda}ph}{m}} + 2\left(\frac{3\mu_2(\nu)\lambda(2p+2)}{r}\right)^{\frac{1}{2}}.$$

This completes the proof of the first claim.

We then proceed to prove the second claim. By the triangle inequality and the monotonicity of Wasserstein distance, we have

$$\begin{aligned}
\mathsf{W}_1(\nu_n^{\mathsf{LMC}}, \nu) &\leqslant \mathsf{W}_1(\nu_n^{\mathsf{LMC}}, \nu^{\lambda}) + \mathsf{W}_1(\nu^{\lambda}, \nu) \\
&\leqslant \mathsf{W}_2(\nu_n^{\mathsf{LMC}}, \nu^{\lambda}) + \mathsf{W}_1(\nu^{\lambda}, \nu).
\end{aligned}$$

When $\nu_0^{\mathsf{LMC}}$ is the Dirac mass at the minimizer of the function $f$, by Proposition 1 in Durmus & Moulines (2019), it holds that

$$\mathsf{W}_2(\nu_0^{\mathsf{LMC}}, \nu^{\lambda}) \leqslant \sqrt{\frac{p}{m}}.$$

Combining this with previous display, display C.2, and Proposition 2.1 then gives

$$\begin{aligned}
\mathsf{W}_1(\nu_n^{\mathsf{LMC}}, \nu) &\leqslant e^{-\frac{mnh}{2}}\mathsf{W}_2(\nu_0^{\mathsf{LMC}}, \nu^{\lambda}) + \sqrt{\frac{2M^{\lambda}ph}{m}} + \mathsf{W}_1(\nu^{\lambda}, \nu) \\
&\leqslant e^{-\frac{mnh}{2}}\sqrt{\frac{p}{m}} + \sqrt{\frac{2M^{\lambda}ph}{m}} + \frac{3\mu_1(\nu)\lambda(2p+1)}{r}
\end{aligned}$$

as desired.

$\square$

*Proof of Theorem 3.2.* The proof employs a similar technique to that used in the proof of Theorem 3.1. Note that $M^{\lambda}h \leqslant 1/10$, by the triangle inequality and Theorem 1 from Yu & Dalalyan (2024), we have

$$\begin{aligned}
\mathsf{W}_2(\nu_n^{\mathsf{pRLMC}}, \nu) &\leqslant \mathsf{W}_2(\nu^{\lambda}, \nu) + \mathsf{W}_2(\nu_n^{\mathsf{pRLMC}}, \nu^{\lambda}) \\
&\leqslant \left(1 + \sqrt{\kappa M^{\lambda}h}\big(0.82(M^{\lambda}h)^{Q-1} + 0.94M^{\lambda}h/R)\big)\right)e^{-mnh/2}\mathsf{W}_2(\nu_0^{\mathsf{pRLMC}}, \nu^{\lambda}) \\
&\quad + \sqrt{\kappa M^{\lambda}h}\big(3.98(M^{\lambda}h)^{Q-1} + 6.91M^{\lambda}h/\sqrt{R}\big)\sqrt{p/m} + \mathsf{W}_2(\nu^{\lambda}, \nu) \\
&\leqslant \left(1 + \sqrt{\kappa M^{\lambda}h}\big(0.82(M^{\lambda}h)^{Q-1} + 0.94M^{\lambda}h/R)\big)\right)e^{-mnh/2}\big(\mathsf{W}_2(\nu_0^{\mathsf{pRLMC}}, \nu) + \mathsf{W}_2(\nu, \nu^{\lambda})\big) \\
&\quad + \sqrt{\kappa M^{\lambda}h}\big(3.98(M^{\lambda}h)^{Q-1} + 6.91M^{\lambda}h/\sqrt{R}\big)\sqrt{p/m} + \mathsf{W}_2(\nu^{\lambda}, \nu).
\end{aligned}$$

Combining this with Proposition 2.1, for any $\lambda \in \left(0, \frac{r}{2(p+2)}\right)$, we obtain

$$\begin{aligned}
\mathsf{W}_2(\nu_n^{\mathsf{pRLMC}}, \nu) &\leqslant \left(1 + \sqrt{\kappa M^{\lambda}h}\big(0.82(M^{\lambda}h)^{Q-1} + 0.94M^{\lambda}h/R)\big)\right)e^{-mnh/2}\mathsf{W}_2(\nu_0^{\mathsf{pRLMC}}, \nu) \\
&\quad + \sqrt{\kappa M^{\lambda}h}\big(3.98(M^{\lambda}h)^{Q-1} + 6.91M^{\lambda}h/\sqrt{R}\big)\sqrt{p/m} \\
&\quad \left(2 + \sqrt{\kappa M^{\lambda}h}\big(0.82(M^{\lambda}h)^{Q-1} + 0.94M^{\lambda}h/R)\big)\right)\sqrt{3\mu_2(\nu)\lambda(2p+2)/r}
\end{aligned}$$

as desired. Invoking Proposition 2.1 and the monotonicity of the Wasserstein distance, since $\nu_0^{\mathsf{pRLMC}}$ is the Dirac mass at the minimizer of the function $f$, it holds that

$$\begin{aligned} \mathsf{W}_1(\nu_n^{\mathsf{pRLMC}}, \nu) &\leqslant \mathsf{W}_2(\nu_n^{\mathsf{pRLMC}}, \nu^\lambda) + \mathsf{W}_1(\nu^\lambda, \nu) \\ &\leqslant \Big(1 + \sqrt{\kappa M^\lambda h}\big(0.82(M^\lambda h)^{Q-1} + 0.94 M^\lambda h/R\big)\Big) e^{-mnh/2}\sqrt{p/m} \\ &\quad + \sqrt{\kappa M^\lambda h}\big(3.98(M^\lambda h)^{Q-1} + 6.91 M^\lambda h/\sqrt{R}\big)\sqrt{p/m} \\ &\quad \Big(2 + \sqrt{\kappa M^\lambda h}\big(0.82(M^\lambda h)^{Q-1} + 0.94 M^\lambda h/R\big)\Big)\sqrt{3\mu_2(\nu)\lambda(2p+2)/r} \end{aligned}$$

as desired. $\qquad\square$

*Proof of Theorem 3.3.* The proof relies a similar technique to that used in the proof of Theorem 3.1. The key distinction lies in the utilization of Theorem 3 from Yu et al. (2023) to bound the $\mathsf{W}_2$ distance between $\nu_n^{\mathsf{KLMC}}$ and $\nu^\lambda$. By the triangle inequality and Theorem 3 from Yu et al. (2023), we have

$$\begin{aligned} \mathsf{W}_2(\nu_n^{\mathsf{KLMC}}, \nu) &\leqslant \mathsf{W}_2(\nu^\lambda, \nu) + \mathsf{W}_2(\nu_n^{\mathsf{KLMC}}, \nu^\lambda) \\ &\leqslant 2\varrho^n \mathsf{W}_2(\nu_0^{\mathsf{KLMC}}, \nu^\lambda) + 0.05\sqrt{\frac{\varrho^n \mathbb{E}[U^\lambda(\boldsymbol{\vartheta}_0^{\mathsf{KLMC}}) - f(\mathbf{0})]}{m}} + 0.9\gamma h\sqrt{\frac{\kappa p}{m}} + \mathsf{W}_2(\nu^\lambda, \nu) \\ &\leqslant 2\varrho^n\big(\mathsf{W}_2(\nu_0^{\mathsf{KLMC}}, \nu) + \mathsf{W}_2(\nu, \nu^\lambda)\big) + 0.05\sqrt{\frac{\varrho^n \mathbb{E}[U^\lambda(\boldsymbol{\vartheta}_0^{\mathsf{KLMC}}) - f(\mathbf{0})]}{m}} + 0.9\gamma h\sqrt{\frac{\kappa p}{m}} \\ &\quad + \mathsf{W}_2(\nu^\lambda, \nu)\,. \end{aligned}$$

Combining this with Proposition 2.1, for any $\lambda \in \big(0, \frac{r}{2(p+2)}\big)$, we obtain

$$\begin{aligned} \mathsf{W}_2(\nu_n^{\mathsf{KLMC}}, \nu) &\leqslant 2\varrho^n \mathsf{W}_2(\nu_0^{\mathsf{KLMC}}, \nu) + 3\sqrt{\frac{3\mu_2(\nu)\lambda(2p+2)}{r}} + 0.05\sqrt{\frac{\varrho^n \mathbb{E}[U^\lambda(\boldsymbol{\vartheta}_0^{\mathsf{KLMC}}) - f(\mathbf{0})]}{m}} \\ &\quad + 0.9\gamma h\sqrt{\frac{\kappa p}{m}} \end{aligned}$$

as desired.

Invoking Proposition 2.1 and the monotonicity of the Wasserstein distance, since $\boldsymbol{\vartheta}_0^{\mathsf{KLMC}} = \mathbf{0}$, it holds that

$$\begin{aligned} \mathsf{W}_1(\nu_n^{\mathsf{KLMC}}, \nu) &\leqslant \mathsf{W}_2(\nu_n^{\mathsf{KLMC}}, \nu^\lambda) + \mathsf{W}_1(\nu^\lambda, \nu) \\ &\leqslant 2\varrho^n\sqrt{\frac{p}{m}} + \frac{3\mu_1(\nu)\lambda(2p+1)}{r} + 0.9\gamma h\sqrt{\frac{\kappa p}{m}} \end{aligned}$$

as desired.

$\qquad\square$

*Proof of Theorem 3.4.* By the triangle inequality and Theorem 2 from Yu & Dalalyan (2024), we have

$$\begin{aligned} \mathsf{W}_2(\nu_n^{\mathsf{pRKLMC}}, \nu) &\leqslant \mathsf{W}_2(\nu^\lambda, \nu) + \mathsf{W}_2(\nu_n^{\mathsf{pRKLMC}}, \nu^\lambda) \\ &\leqslant 1.8\varrho^n \mathsf{W}_2(\nu_0^{\mathsf{pRKLMC}}, \nu^\lambda) + 0.28\sqrt{\frac{\varrho^n \mathbb{E}[U^\lambda(\boldsymbol{\vartheta}_0^{\mathsf{pRKLMC}}) - f(\mathbf{0})]}{m}} \\ &\quad + 44.78\sqrt{\frac{(\gamma M^\lambda h^2)^3}{R^2} + (\gamma M^\lambda h^2)^{2Q-1}}\sqrt{\frac{\kappa p}{m}} + \mathsf{W}_2(\nu^\lambda, \nu) \\ &\leqslant 1.8\varrho^n\big(\mathsf{W}_2(\nu_0^{\mathsf{pRKLMC}}, \nu) + \mathsf{W}_2(\nu, \nu^\lambda)\big) + 0.28\sqrt{\frac{\varrho^n \mathbb{E}[U^\lambda(\boldsymbol{\vartheta}_0^{\mathsf{pRKLMC}}) - f(\mathbf{0})]}{m}} \\ &\quad + 44.78\sqrt{\frac{(\gamma M^\lambda h^2)^3}{R^2} + (\gamma M^\lambda h^2)^{2Q-1}}\sqrt{\frac{\kappa p}{m}} + \mathsf{W}_2(\nu^\lambda, \nu)\,. \end{aligned}$$

Combining this with Proposition 2.1, for any $\lambda \in \left(0, \frac{r}{2(p+2)}\right)$, we obtain

$$W_2(\nu_n^{\mathsf{pRKLMC}}, \nu) \leqslant 1.8\varrho^n W_2(\nu_0^{\mathsf{pRKLMC}}, \nu) + 0.28\sqrt{\frac{\varrho^n \mathbb{E}[U^\lambda(\boldsymbol{\vartheta}_0^{\mathsf{pRKLMC}}) - f(\mathbf{0})]}{m}}$$
$$+ 44.78\sqrt{\frac{(\gamma M^\lambda h^2)^3}{R^2} + (\gamma M^\lambda h^2)^{2Q-1}}\sqrt{\frac{\kappa p}{m}} + 2.8\sqrt{\frac{3\mu_2(\nu)\lambda(2p+2)}{r}}$$

as desired.

Invoking Proposition 2.1 and the monotonicity of the Wasserstein distance, since $\boldsymbol{\vartheta}_0^{\mathsf{pRKLMC}} = \mathbf{0}$, it holds that

$$W_1(\nu_n^{\mathsf{pRKLMC}}, \nu) \leqslant W_2(\nu_n^{\mathsf{pRKLMC}}, \nu^\lambda) + W_1(\nu^\lambda, \nu)$$
$$\leqslant 1.8\varrho^n\sqrt{\frac{p}{m}} + \frac{3\mu_1(\nu)\lambda(2p+1)}{r} + 44.78\sqrt{\frac{(\gamma M^\lambda h^2)^3}{R^2} + (\gamma M^\lambda h^2)^{2Q-1}}\sqrt{\frac{\kappa p}{m}}$$

as desired.

$\square$