# OpenReview forum: "Log-Concave Sampling on Compact Supports: A Versatile Proximal Framework"
_ICLR.cc/2025/Conference — ICLR 2025 Conference Withdrawn Submission_

### Official Review · Reviewer_ydeo · 2024-10-31

**Soundness:** 3
**Presentation:** 2
**Contribution:** 1
**Rating:** 3
**Confidence:** 4

**Summary:**

This paper investigate the problem of sampling from strongly log-concave distributions on convex and compact supports. This paper provides upper bound on the $W_1$ and $W_2$ errors. The key point is to upper bound the Wasserstein distance between the indicator function of support set and its approximation version, which is the only contribution in their work.

**Strengths:**

- The main contribution is to upper bound the Wasserstein distance between the indicator function of support set and its approximation version, which is tight for $W_1$ distance.
- This paper further applies several existing methods and obtain several upper bounds for constrained log-concave sampling w.r.t. $W_1$ and $W_2$ distances.

**Weaknesses:**

- The accuracy assessment of the approximation is limited to the Wasserstein distance, which restricts the scope of evaluation.
- This work appears to be a straightforward combination of a novel approximation approach with existing log-concave methods.
- The presentation of the complexity analysis is unclear. I do not find the statements about the query complexity of the gradient oracle and the membership oracle.

**Questions:**

- Is there a possibility to extend your approximation bounds to other metrics, such as TV and KL divergence? A comprehensive analysis of approximation under various metrics would be highly beneficial for non-smooth log-concave sampling.
- Can you compare your smoothing methods with other commonly used techniques, such as  l_2/l_1  ball smoothing [1] and Gaussian smoothing [2]?
- Can you provide a comparison of your upper bounds with existing upper and lower bounds for constrained log-concave sampling, considering the complexity of the gradient oracle of  $f$  and the membership oracle of  $K$ , as in [3]? Additionally, generally what are the main advantages of using a smoothing-based method? Furthermore, in your setting, where the non-smooth part is primarily on the boundary, I think bounding the Lipschitz constant of the gradient in general could lead to poor complexity results.

[1] Randomized gradient-free methods in convex optimization, A Gasnikov, D Dvinskikh, P Dvurechensky, E Gorbunov, A Beznosikov, A Lobanov, 2023

[2] Langevin monte carlo without smoothness, Niladri Chatterji, Jelena Diakonikolas, Michael I. Jordan, Peter Bartlett, 2020

[3] In-and-Out: Algorithmic Diffusion for Sampling Convex Bodies Yunbum Kook, Santosh S. Vempala, Matthew S. Zhang, 2024

---

### Official Review · Reviewer_85dY · 2024-11-04

**Soundness:** 3
**Presentation:** 2
**Contribution:** 2
**Rating:** 5
**Confidence:** 3

**Summary:**

This paper considers the problem of sampling from a logconcave distribution, where the log-density is strongly convex and smooth, and is constrained to a compact convex set $K$.  The paper analyzes different verisions of the Langevin algorithm (Vanilla Langevin, Kinetic Langevin, as well as randomized midpoint method implementation of these algorithms).  The constraints are enforced via one of two different projections onto the convex set $K$:  the proximal projection which requires access to a projection oracle for the convex constraint set $K$, and the Gauge projection which can be computed with access to a membership oracle for the convex constraint set $K$.  The paper provides runtime bounds for each of these algorithms when sampling within a specified Wasserstein-1 or Wasserstein-2 error $\epsilon$ of the target distribution.

**Strengths:**

The paper shows new bounds for different Langevin sampling algorithms, when sampling from a constrained logconcave distribution with strongly convex and smooth log-density.  The fact that the distribution is constrained makes it difficult to implement first-order (and higher-order) algorithms like Langevin Monte Carlo, requiring the authors' use of a proximal or Gauge operator.  The authors provide runtime bounds for three different versions of the Langevin sampling algorithm, and they do so in both W-1 and W-2 distance.

**Weaknesses:**

The main weakness of the paper is that it does not seem to improve much on the runtime over previous works for the problem of sampling from constrained logconcave distributions with strongly convex and smooth log-density considered in the paper.  It analyzes different versions of the Langevin algorithm applied to this sampling problem, but does not say whether it improves over the runtime bounds for the best algorithm for this problem.

It would also be helpful to better explain the novelty of the technical contribution in the introduction, and to better explain what proof/algorithmic techniques are new and which are borrowed from previous works  (perhaps this could be explained in an "our contributions subsection" in the introduction,  or in a technical overview section, as it is currently difficult to assess the novelty of the techniques.

**Questions:**

It would be helpful if the authors could clarify whether and to what extent their results improve over the runtime bounds of prior works (both in comparison to prior bounds for the Langevin algorithm and prior bounds for other algorithms) for the sampling problem considered in this work.  A table with the comparisons might be helpful here as well.

I would also be helpful to explain the novelty of the technical contributions in the introduction (perhaps in an "our contributions" subsection of the introduction).

---

### Official Review · Reviewer_1yKV · 2024-11-04

**Soundness:** 4
**Presentation:** 3
**Contribution:** 3
**Rating:** 3
**Confidence:** 2

**Summary:**

The paper proposes a new method for sampling probability distributions defined over convex sets by approximating the density with a surrogate distribution that is smooth and strongly convex, such that standard sampling and projections can be done. Their focus is on providing theoretical insights, which is also apparently why they provide zero simulated or real data experiments of any kind.

They use a surrogate potential U that is the original potential function f + a softener $\ell$ like the Moreau envelope.


Main stated contributions are giving upper and lower bounds between their proposed approximation and the target density in generalized Wasserstein distance, and what they think is the first convergence rate for the parallelized pRLMC algorithm for constrained sampling.

**Strengths:**

This is a fun problem.

**Weaknesses:**

Motivation: The authors repeatedly tell us how important this work is, for example, they tell us this work:  “provides valuable theoretical insights” “enables clear comparisons” “exceptionally flexible and adaptable.” But as they say in cinema, “show don’t tell”.  Because by the end of page 2 I was still looking for at least one motivating example. I flipped to the experiments section to hopefully see how useful this would be, but came up dry, as this is solely a theory paper.  I can come up with a couple cases I’ve run into for sampling from constrained distributions, but nothing strong enough to argue for acceptance of a theory paper this narrow. Please authors / other-reviewers, help me out and argue for why this is actually an important problem that needs solving by the ICLR community?  If ICLR accepted 50% of the papers, motivation wouldn’t be such an issue and I'd be more positive just on the scholarship, but with the low acceptance rate of ICLR, we need to have a strong reason that the ICLR community should be bumping other good papers to publish this one. I am very open to raising my score if (especially) the other reviewers or Area Chair can motivate why this is an important paper to publish *at ICLR*.

MORE FEEDBACK:

Explain Assumption 2.1 better please when you introduce it. I’m not familiar enough with that related work to know why assumption 2.1 is a needed assumption, and WOLOG (without loss of generality) I would have expected it to not be a particularly big deal that I can fit *some* ball inside the density perhaps with a very tiny $r$, but maybe that’s all you do mean because all you need is to be able to have some $r > 0$ for Proposition 2.1?  OR is it that Assumption 2.1 says “Given a positive constant $r$, so do I get to choose any $r$ and then you’re saying the convex set $\mathcal{K}$ will contain the $r$-ball? That seems so weak as to be useless. But maybe the implication here is that it is WOLOG, and you could stretch $\mathcal{K}$ out to contain the $r$-ball WOLOG, which is essentially saying just some tiny ball around $0$ has to fit inside the set $\mathcal{K}$?  I’d find it really helpful if you can foreshadow/clarify a bit better what this assumption really is intended to mean and why.  I note later you note this is an important differentiator from Brosse 2017 which assumes the domain is containable with a ball, but is this really a substantive win? Can you give some real-world / practical applications/examples where Brosse’s assumption wouldn’t hold but the less-assumptive 2.1 does?

Assumption 2.2 While I agree this is a standard assumption, kindly define $m$ and $M$ for the record (e.g. by adding “there exists some $m$ and $M$ such that”).

MINOR COMMENTS:

TYPO: “we the define the”

GRAMMAR: “set \mathcal{D} satisfy” —> “satisfies”

Remember to use {Langevin}, {Monte Carlo} etc. in your bib text to get proper capitalization in paper tiles in your references.

**Questions:**

Why is this significant? Why is this especially significant and or interest for the ICLR community?

---

### Note · Authors · 2024-11-21

I have read and agree with the venue's withdrawal policy on behalf of myself and my co-authors.